# SwiFT: Swin 4D fMRI Transformer

**Peter Yongho Kim**[*]
Seoul National University

**Junbeom Kwon**[*]
Seoul National University

**Sunghwan Joo**
SungKyunKwan University

**Sangyoon Bae**
Seoul National University

**Donggyu Lee**
SungKyunKwan University

**Yoonho Jung**
Seoul National University

**Shinjae Yoo**
Brookhaven National Lab

**Jiook Cha** [†]
Seoul National University

**Taesup Moon** [†]
Seoul National University

## Abstract

Modeling spatiotemporal brain dynamics from high-dimensional data, such as functional Magnetic Resonance Imaging (fMRI), is a formidable task in neuroscience. Existing approaches for fMRI analysis utilize hand-crafted features, but the process of feature extraction risks losing essential information in fMRI scans. To address this challenge, we present SwiFT (**Swi**n 4D **f**MRI **T**ransformer), a Swin Transformer architecture that can learn brain dynamics directly from fMRI volumes in a memory and computation-efficient manner. SwiFT achieves this by implementing a 4D window multi-head self-attention mechanism and absolute positional embeddings. We evaluate SwiFT using multiple large-scale resting-state fMRI datasets, including the Human Connectome Project (HCP), Adolescent Brain Cognitive Development (ABCD), and UK Biobank (UKB) datasets, to predict sex, age, and cognitive intelligence. Our experimental outcomes reveal that SwiFT consistently outperforms recent state-of-the-art models. Furthermore, by leveraging its end-to-end learning capability, we show that contrastive loss-based self-supervised pre-training of SwiFT can enhance performance on downstream tasks. Additionally, we employ an explainable AI method to identify the brain regions associated with sex classification. To our knowledge, SwiFT is the first Swin Transformer architecture to process dimensional spatiotemporal brain functional data in an end-to-end fashion. Our work holds substantial potential in facilitating scalable learning of functional brain imaging in neuroscience research by reducing the hurdles associated with applying Transformer models to high-dimensional fMRI. Project page: https://github.com/Transconnectome/SwiFT

## 1 Introduction

The human brain is a dynamic system characterized as an extensive (feedback) network generating complex spatiotemporal dynamics of its activity. Analyzing such dynamics and linking them to normative adaptive cognition and behaviors, as well as their maladaptive forms in a disease condition is of paramount importance in neuroscience and medicine. However, the fields have been hampered by a lack of predictive power owing to the gap between the complexity of the brain network and the contrasting simplicity of brain imaging analytics [1]. Hence, developing an accurate predictive model using high-dimensional brain imaging data and learning rich representations of brain function will help close this gap and advance toward precision neuroscience.

---

[*]Equal Contribution
[†]Co-corresponding author

37th Conference on Neural Information Processing Systems (NeurIPS 2023).

Among many brain imaging modalities, functional Magnetic Resonance Imaging (fMRI) provides a unique opportunity to model the spatiotemporal patterns of brain electrochemical activity. Functional MRI captures a temporal sequence of 3D images (a stack of 2D slices) of Blood Oxygenation Level Dependent (BOLD) signals with time resolution ranging from 0.5 to 3 seconds, rendering in 4D data. It has accelerated the discovery of detailed functional anatomy of the human brain and significantly contributed to the understanding of brain diseases, such as Alzheimer's disease and psychiatric disorders. To better process the fMRI data, statistical methods were conventionally used to estimate the spatiotemporal patterns related to cognition [2, 3] and brain diseases [4, 5, 6, 7]. More recently, deep neural networks (DNN) have been applied to fMRI to investigate the nonlinear relationship of brain dynamics with human cognition and behaviors [8, 9, 10, 11, 12].

Regarding the DNN-based approaches, researchers have broadly pursued two distinct lines of work. The first approach is the so-called *ROI-based method*, in which the high-dimensional fMRI data (with around 300,000 voxels) is clustered into the temporal sequence of hundreds of pre-defined brain regions (ROIs) using anatomical segmentation [13] or statistical clustering [14] before learning a predictive DNN model. Despite the computational efficiency, these pre-processing approaches may be prone to losing information important to capture subtle variability across the individual brains [15]. The second DNN-based approach is the *two-step method*, in which the raw fMRI data is used as input, and specialized architectures for spatial and temporal domains are separately used. Namely, for learning spatial features, convolutional neural networks (CNNs) are used, and for temporal, long short-term memory (LSTM) [16] or Transformers [17, 18] are used. This separation of DNN architecture necessarily results in two-step learning for better computational and memory efficiency. However, it could also limit the capability of capturing comprehensive information among brain regions across the temporal dimension. Therefore, a critical unresolved issue is whether an efficient, end-to-end DNN that utilizes raw 4D fMRI input can be formulated to better model and learn the brain dynamics compared to previous approaches.

To that end, we propose **Swi**n 4D **f**MRI **T**ransformer **(SwiFT)**, a 4D extension of the Swin Transformer [19] architecture, which can jointly learn the spatiotemporal representations of the brain's intrinsic activity directly from high-dimensional fMRI in an end-to-end fashion. The main gist of our method is to employ the 4D local window attention structure, which makes SwiFT readily applicable to process large-scale, high-dimensional 4D data with low computational complexity. We note that while 3D variants of Swin Transformers have been proposed before [20, 21, 22, 23] to take video or medical image inputs, to the best of our knowledge, this is the first work to extend Swin Transformer to take 4D data input and to apply it to the fMRI data.

Our experimental results show that the end-to-end learning capability of SwiFT unlocks its potential to effectively learn complex spatiotemporal patterns in fMRI. Specifically, we evaluate SwiFT's performance on three representative fMRI benchmarks: the Human Connectome Project (HCP) [24], the Adolescent Brain Cognitive Development (ABCD) [25], and the UK Biobank (UKB) [26, 27]. Across various classification and regression tasks, including sex classification and age/intelligence prediction, SwiFT significantly outperforms the recent baselines mentioned above. Furthermore, we also demonstrate that applying the widely successful "pre-train and fine-tune" framework to SwiFT would be feasible. We believe this capability has the potential to empower researchers to construct large-scale foundation models for fMRI, akin to those utilized in several other application domains. Finally, to provide a comprehensive analysis, we present the interpretation results using IG-SQ for SwiFT's predictions and conduct ablation studies to substantiate our modeling choices.

## 2 Related Work

**ROI-based methods** To analyze the brain network, researchers typically reduce the sequences of brain volumes into low-dimensional multivariate time series by aggregating voxel intensities in specific regions of interest (ROI) based on standardized brain atlases, considering pairwise correlations between the time series of each ROI as functional connectivity [28, 29]. The majority of the DNNs, such as graph neural networks (GNN), were designed to treat the brain network as a graph, considering each ROI as node and pairwise correlations between the ROIs as edges. For instance, BrainNetCNN, consisting of multiple graph convolutional filters, was proposed to model various levels of topological interactions in structural [30] and functional connectivity [12]. Kan et al. [12] proposed a Transformer-based model that learns the individual connectivity strengths between each ROI. Namely, they apply an orthonormal clustering readout operation for functional

connectivity to locate functional clusters related to specific human behaviors, and those clusters serve as informative embeddings for predicting psychiatric and biological outcomes. Some recent studies proposed methods to capture spatiotemporal dynamics directly from extracted fMRI time series, utilizing Transformer by separating spatial and temporal attention units [31] and focusing on local representations with fused window multi-head self-attention (FW-MSA) [32].

**Two-step methods for 4D fMRI inputs**    Existing DNNs that take raw 4D fMRI as input typically process spatial and temporal information separately. Li et al. [16] integrated 3D convolutional neural networks (CNNs) to extract spatial embeddings in each 3D volume of a 4D fMRI and then feed the spatial embeddings to an LSTM for temporal encoding. Transformer Framework for fMRI (TFF) by Malkiel et al. [17] replaces the LSTM with a Transformer to extract temporal features and propose reconstruction-based pre-training steps. To learn spatial features before the downstream task, TFF concatenates decoder layers after the Transformer and minimizes three reconstruction-based losses; L1, perceptual, and intensity losses. Nguyen et al. [18] suggested a pre-training method to predict the types of cognitive tasks performed during an fMRI scanning. A 3D CNN encoder is then trained to predict the target variable from fMRI volumes. Furthermore, the pre-trained encoder learns temporal features by attaching multi-head self-attention layers. The aforementioned models may have issues of unstable training of spatiotemporal data, which involves multiple training steps and large memory usage. These limitations may result in sub-optimal model computation and learning capability.

**Transformers for vision tasks**    Following the success of Transformers in natural language processing [33, 34], many works applied the Transformer architecture in vision tasks. One of the major challenges for such an extension is the computation complexity scaling quadratically with respect to the number of tokens. Namely, as images have a much larger number of tokens (pixels), rendering the direct pixel-to-token use of Transformers infeasible in most cases. Dosovitskiy et al. [35] tackled this issue with ViT, converting image patches into tokens rather than treating each individual pixel as a token, but it still did not resolve the quadratic scaling of the self-attention layers. Liu et al. [19] suggested the Swin Transformer, a model that reduces the computational complexity to be linear in the number of tokens by running self-attention only within a local window instead of running it globally. Along with the local windowed attention, Swin Transformer also introduces shifting windows to add cross-window connections and patch merging (downsampling) steps to produce hierarchical representations. Swin UNETR [36] extended the Swin Transformer to handle 3D images and demonstrated its utility by coupling a Swin Transformer encoder with CNN-based decoders. VAT [37], a 4D Convolutional Swin Transformer, extended the Swin Transformer model to accept a 4D correlation map of two CNN-extracted 2D image features, utilizing the model for cost aggregation. While these works show that the Swin Transformer model can be extended to handle more spatial dimensions, Video Swin Transformer [23] shows that the Swin Transformer model can also be extended to the temporal dimension by processing 3D video inputs. Overall, these works present the feasibility of applying Swin Transformers to higher spatiotemporal dimensions, but to the best of our knowledge, such a method has yet to be applied to 4D functional brain imaging.

## 3    Main Method

### 3.1    Swin 4D fMRI Transformer (SwiFT)

**Overall architecture**    In line with the recent studies [23, 20], which introduce 3D extensions to enhance the capabilities of the Swin Transformer [19], we propose an advancement in the architecture to add an additional temporal dimension, thereby enabling its application to 4D data.

The overall architecture of our model is depicted in Figure 1a. The SwiFT architecture utilized in our study consists of four distinct stages. Each stage is constructed through the implementation of patch merging, with linear embedding employed in the case of Stage 1. Additionally, positional embedding is incorporated, and multiple (Swin) Transformer blocks are applied repeatedly within the stages. The model processes an input fMRI data of size $T \times H \times W \times D \times 1$, which consists of a length $T$ sequence of fMRI volumes ($H \times W \times D$) with a single channel. During the initial patch partitioning step, the input fMRI data is partitioned into $T \times \frac{H}{P} \times \frac{W}{P} \times \frac{D}{P}$ patches with $P^3$ voxels. In this study, $H$, $W$, and $D$ are 96, and the initial patch size $P$ is 6. During the linear embedding process, patches with size $P^3$ are transformed into $C$-dimensional tokens. This transformation effectively maps the spatially neighboring pixels within a patch onto a token.

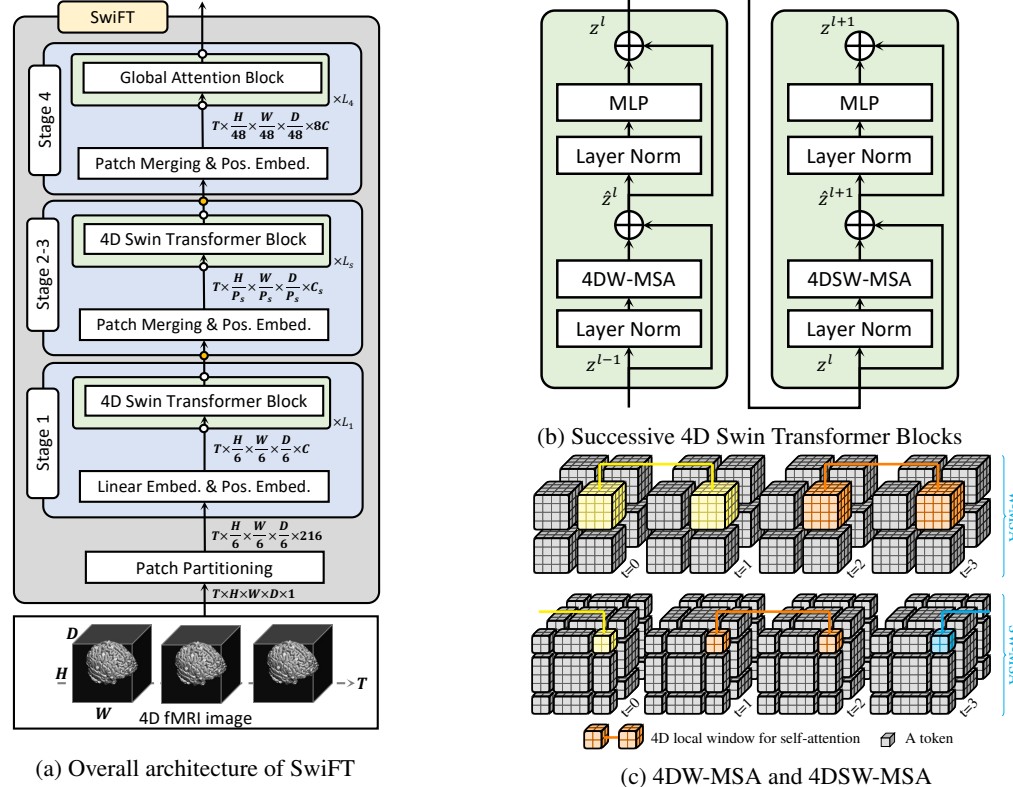

(a) Overall architecture of SwiFT

(b) Successive 4D Swin Transformer Blocks

(c) 4DW-MSA and 4DSW-MSA

Figure 1: Figures depicting the structure of SwiFT and its components.

Next, following an absolute positional embedding layer, multiple layers of 4D Swin Transformer blocks are applied on the embedded patches, forming Stage 1 of SwiFT. Starting from Stage 2, a patch merging layer at the beginning of each stage reduces the number of tokens by merging 8 spatially neighboring patches. After the patch merging layer, an absolute positional embedding layer followed by multiple layers of 4D Swin Transformer blocks is applied, forming Stage 2 and onward. In the final stage, Stage 4, the 4D Swin Transformer blocks are replaced by global attention Transformer blocks which carry out global attention instead of local window attention. This computationally expensive global attention is made possible by the significant reduction in the number of tokens achieved through the patch merging steps executed in the preceding stages. Global attention Transformer blocks allow each token to globally attend to all other tokens rather than being restricted to the tokens within the local window.

**Patch merging**  Following prior works [23, 20, 38, 39], the patch merging step is only performed for the three spatial dimensions $(H, W, D)$ and not for the temporal dimension $(T)$, thereby merging a group of $8 = 2 \times 2 \times 2$ neighboring patches into a single patch for each time frame. During the patch merging operation, the spatial dimensions are reduced by half, and the channel size $(C)$ is doubled as compensation. Thus, in Figure 1a, the numbers $P_2, P_3, C_2,$ and $C_3$ are $12, 24, 2C,$ and $4C$, respectively.

As a general example, consider a tensor with arbitrary dimensions $T \times H' \times W' \times D' \times C'$ before passing through the patch merging layer. During patch merging, this tensor is reshaped into a new tensor with dimensions $T \times \frac{H'}{2} \times \frac{W'}{2} \times \frac{D'}{2} \times 8C'$, where $2 \times 2 \times 2$ spatially neighboring patches are concatenated along the channel dimension. Then, each $8C'$ channel in the resulting tensor is projected onto a $2C'$ dimensional space by applying a single fully connected layer, resulting $T \times \frac{H'}{2} \times \frac{W'}{2} \times \frac{D'}{2} \times 2C'$ in total. The process of patch merging facilitates the hierarchical feature-extraction structure of SwiFT and reduces the computational complexity of the subsequent layers. We clarify that while the patch merging is operated only on the spatial dimensions, the temporal information is still well-incorporated via the windowed attention.

**4D window multi-head self-attention** The core of the Swin Transformer model is the window multi-head self-attention (W-MSA) layer, which allows the model to process a large number of tokens by limiting self-attention only within a local window. In SwiFT, the 3D window mechanism is extended to 4D windows; given $T \times H' \times W' \times D'$ input tokens, the tokens are partitioned by a window of size $P \times M \times M \times M$, resulting in $\lceil \frac{T}{P} \rceil \times \lceil \frac{H'}{M} \rceil \times \lceil \frac{W'}{M} \rceil \times \lceil \frac{D'}{M} \rceil$ non-overlapping windows.

However, simply stacking multiple window self-attention layers would be undesirable since there would be no crosstalk across different windows. To that end, a shifted window multi-head self-attention (SW-MSA) layer enables cross-window connections. Namely, we extend the 3D shifted window mechanism to 4D shifted windows as well; in $P \times M \times M \times M$ windows obtained from the W-MSA layer, we shift the windows of the successive layer by $(\frac{P}{2}, \frac{M}{2}, \frac{M}{2}, \frac{M}{2})$ tokens.

The detailed operations of our 4D W-MSA and SW-MSA are shown in Figure 1c. In this example, the applied size of input tokens and the windows are $T \times H' \times W' \times D' = 4 \times 8 \times 8 \times 8$ and $P \times M \times M \times M = 2 \times 4 \times 4 \times 4$, respectively. Then, by following the window partitioning methods described above, the numbers of grouped windows in W-MSA and SW-MSA become $2 \times 2 \times 2 \times 2 = 16$ and $3 \times 3 \times 3 \times 3 = 81$, respectively. Such separately applied window self-attention is essential in effectively extracting spatiotemporal feature representation from the 4D fMRI data. Although the number of windows increases in SW-MSA, the actual computation cost remains similar by leveraging the cyclic-shifting batch computation proposed in [19].

Combining the W-MSA layer and the SW-MSA layer, two successive 4D Swin Transformer blocks, as shown in Figure 1b, are computed as the following:

$$\hat{z}^l = \text{4DW-MSA}(\text{LN}(z^{l-1})) + z^{l-1}, \quad z^l = \text{MLP}(\text{LN}(\hat{z}^l)) + \hat{z}^l$$
$$\hat{z}^{l+1} = \text{4DSW-MSA}(\text{LN}(z^l)) + z^l, \quad z^{l+1} = \text{MLP}(\text{LN}(\hat{z}^{l+1})) + \hat{z}^{l+1},$$

in which 4D(S)W-MSA, LN, and MLP denote the 4D (Shifted) Window Multi-head Self-Attention, Layer Normalization, and Multi-Layer Perceptron module, respectively. Moreover, $\hat{z}^l$ and $z^l$ denote the output features of the 4D(S)W-MSA module and the following MLP module for block $l$, respectively.

**4D absolute positional embedding** Even though previous models utilize relative position biases to encode positional information [19, 23], we have opted instead for an absolute position embedding scheme for SwiFT. While absolute positional embeddings are more computationally expensive for low-dimensional data [19], since we are dealing with much larger scale 4D data, absolute positional embeddings become more cost-effective than relative positional bias. To that end, we add a learnable embedding at the beginning of each stage of the Transformer right after the patch merging step. In line with [22], we separately add positional embeddings for the spatial and temporal dimensions. Specifically, given an input tensor with dimensions of $T \times H' \times W' \times D' \times C'$, we define spatial and temporal positional embedding tensors with dimensions of $1 \times H' \times W' \times D' \times C'$ and $T \times 1 \times 1 \times 1 \times C'$, respectively. These tensors are then added to the input tensor using broadcasting. The effect of this switch is investigated in Appendix C.1, and overall absolute positional embedding seems to be the superior method of choice.

## 3.2 Self-supervised Pre-training

Our proposed end-to-end model structure allows efficient self-supervised pre-training of SwiFT, which can then be fine-tuned for specific tasks. This unique capability sets our method apart from other approaches in Section 2 relying on ROI-based brain data or a two-step learning approach for 4D fMRI. We achieve this by using two different contrastive loss-based pre-training objectives adapted from [40]. Figure 2 depicts the positive and negative pairs for the two loss functions, where the InfoNCE [41] loss of the pairs is calculated for the final loss function.

**Instance contrastive loss** To allow the model to distinguish fMRI scans that come from *different subjects*, the instance contrastive loss ($\mathcal{L}_{IC}$) is a type of contrastive-based loss function that considers a representation to be positive if two distinct fMRI sub-sequences come from the same subject and negative if they come from different subjects. The feature representation passes through three layers: SwiFT, global average pooling, and a multi-layer perceptron (MLP) head. To clearly define this loss function, we denote the feature representation as $f_{i,p}$, where $i \in \{1, ..., B\}$ refers to the subject index for a given batch size $B$, and $p$ refers to the fMRI sub-sequence index (either 1 or 2). Since we are

sampling two sub-sequences for each of the $B$ subjects, this amounts to a total of $2B$ representations. For each subject $i$, the anchor, positive, and negative representations are set as $f_{i,1}$, $f_{i,2}$, and the remaining $2B - 2$ feature representations, respectively. Using this setup, the instance contrastive loss for subject $i$ is denoted and defined as

$$\mathcal{L}_{IC}^i = - \log \frac{h(f_{i,1}, f_{i,2})}{\sum_{j=1}^{B} [\mathbb{1}_{[j \neq i]} (h(f_{i,1}, f_{j,1}) + h(f_{i,1}, f_{j,2}))]},$$

in which $h$ denotes the exponential of the cosine similarity between two vectors, and $\mathbb{1}$ denotes an indicator function that equals one if the condition is true and zero otherwise.

**Local-local temporal contrastive loss**    To allow the model to distinguish fMRI scans that come from *different timestamps within the same subject*, the local-local temporal contrastive loss ($\mathcal{L}_{LL}$) determines both positive and negative pairs within a single subject. Namely, a positive representation is derived from the same fMRI sub-sequence but is applied with different random augmentations. On the other hand, negative representations come from distinct fMRI sub-sequences from the same subjects. The feature representations are obtained in the same manner as the instance contrastive loss. To clearly define this loss function, we use the same notation of feature representation as $f_{i,p}$, but the range of $p$ is changed to $\{1, 2, ..., N\}$, where $N$ is the number of fMRI sub-sequences from a single subject. We denote an fMRI sub-sequence of the same subject $i$ and fMRI sub-sequence $p$ but different random augmentation as $\tilde{f}_{i,p}$. Since we are sampling two differently augmented versions for each of the $N$ sub-sequences, this amounts to a total of $2N$ representations. Using this setup, the local-local temporal contrastive loss for subject $i$ is denoted as $\mathcal{L}_{LL}^i$ and defined as

$$\mathcal{L}_{LL}^i = - \sum_{p=1}^{N} \log \frac{h(f_{i,p}, \tilde{f}_{i,p})}{\sum_{q=1}^{N} [\mathbb{1}_{[q \neq p]} (h(f_{i,p}, f_{i,q}) + h(f_{i,p}, \tilde{f}_{i,q}))]},$$

in which $h$ denotes the exponential of the cosine similarity between two vectors.

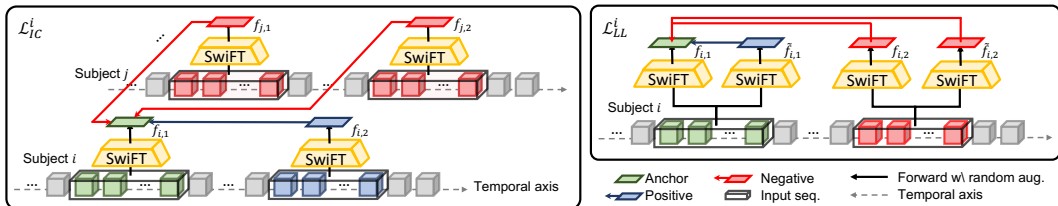

Figure 2: Illustration of two different contrastive losses for the pre-training of SwiFT.

## 4 Experiments

### 4.1 Experimental Settings

**Datasets**    The Adolescent Brain Cognitive Development (ABCD) study is a longitudinal, multi-site investigation of brain development and related behavioral outcomes in children [25]. The dataset is open to the scientific community but requires authorization. After quality control, we used the resting state fMRI of 9,128 children (age $= 118.95 \pm 7.46$ months, 52.4% female) from release 2. For fMRI preprocessing, we used a well-established pipeline, fMRIprep [42, 43], which includes reducing the bias field, skull-stripping, alignment to structural image, and spatial normalization to standard space for a pediatric brain [44]. After fMRIprep, we applied low pass filtering, head movement correction, and artifact removal, regressing out signals from non-grey matters (aCompcor) [45].

We also used the resting-state fMRI of 1,084 healthy young adults (age $= 28.80 \pm 3.70$ years, 54.4% female) from the Human Connectome Project (HCP) (S1200 data) [46, 47], and 5,935 middle and old aged adults (age $= 54.971 \pm 7.53$ years) from the UK Biobank (UKB) [48]. We used preprocessed data provided by Human Connectome Project [49] and UK Biobank [26, 27], which follows the fMRI volume pipeline, including reducing the bias field, skull-stripping, cross-modality registration, and spatial normalization to standard space.

For each of the 4D fMRI volumes, we globally normalized brain images over the four dimensions except for the background regions. Then we filled the background with the minimal voxel intensity value. To easily divide the volume into patches for SwiFT, we changed the 3D volume into a shape of $96 \times 96 \times 96$ by cropping and padding on the background. To evaluate the performances of ROI-based models, following the preprocessing steps of [12] as closely as possible, we applied the HCP MMP1 atlas [50] to each fMRI volume to obtain the time series data for each ROI. Subsequently, we processed this ROI series to generate functional connectivities, which involves computing the Pearson correlation coefficient to construct a correlation matrix. The correlation matrix is then Fisher Transformed, serving as the input for the ROI-based models in Section 4.1.2.

To evaluate our models, we constructed three random splits with a ratio of (train: validation: test) = $(0.7 : 0.15 : 0.15)$ and reported the average performances across the three splits.

**Targets** We chose the sex [51], age [52], and cognitive intelligence (NIH Toolbox [53] for ABCD, HCP datasets, and "fluid intelligence" for UKB dataset) of each subject as the prediction target for our models. These targets are significant since the relationship between the brain and these targets represents a fundamental brain-biology and brain-cognition association. Also, the capability to predict these outcomes can prove the model's capability to process fMRI volumes, possibly leading to the prediction of clinical outcomes of debilitating brain disorders, such as Alzheimer's disease, schizophrenia, autism, and bipolar disorder [54, 55, 56, 57]. For these reasons, predicting these outcomes from brain imaging has been an important benchmark task in recent computational neuroscience [12, 8, 58].

The regression targets (age, intelligence) were z-normalized to bring stable training regardless of the range of the target variable. Since the age has a unit (e.g., years or months), when reporting the performance metrics, we transformed the z-scaled age back to its original scale of months or years.

For the binary classification task, balanced accuracy and AUC (Area Under ROC Curve) were used to evaluate model performances. For the regression tasks, Mean Squared Error (MSE) and Mean Absolute Error (MAE) were used to evaluate model performances.

### 4.1.1 Implementation Details

For SwiFT, we use the same architecture across all of our experiments, using the architecture corresponding to the Swin Transformer-variant from previous work [19, 23] with a channel number of $C = 36$. The numbers of layers are fixed to $\{L_1, L_2, L_3, L_4\} = \{2, 2, 6, 2\}$ which corresponds to a model with three stages with 2, 2, 6 consecutive 4D Swin Transformer blocks for each stage and a final stage with 2 consecutive global attention Transformer blocks. In the cases of 4D(S)W-MSA, we set $P = M = 4$. The final output of the model is obtained by applying a global average pooling layer on the output feature map of Stage 4, followed by an MLP head. For training, the Binary Cross Entropy (BCE) loss was used for the binary classification task, and the Mean Squared Error (MSE) loss was used for regression tasks.

For the ABCD dataset, input training images were randomly augmented to prevent the model from overfitting. The augmentations include affine transformation, adding Gaussian noise, and Gaussian smoothing, and were also applied for the contrastive pre-training in Section 4.3.

Instead of inputting the entire fMRI volume of a subject, we divided the volume into 20-frame subsequences and used them as the input, mainly due to memory constraints. However, we believe this choice may benefit the model, as discussed in Appendix C.2 in detail. For training, each individual sub-sequence was treated as a data point, meaning the appropriate loss function was calculated and backpropagated for each sub-sequence. For inference, the logits from the sub-sequences of each particular subject were averaged, yielding a single output for each subject.

**Computational complexity** The computational complexities of a single global attention Transformer block (denoted as MSA & MLP) and a 4D Swin Transformer block (denoted as W-MSA & MLP) for input with a dimension of $T \times H' \times W' \times D' \times C'$ can be calculated as

$$\Omega(\text{MSA \& MLP}) = 12NC'^2 + 2N^2C' \quad \Omega(\text{W-MSA \& MLP}) = 12NC'^2 + 2PM^3NC',$$

in which the number of tokens $N = TH'W'D'$. In practice, on Stage 1 of SwiFT, setting the values used for the experiments $C' = 36, T = 20, H' = W' = D' = 16, P = M = 4$, the two terms $12NC'^2$ and $2PM^3NC'$ are balanced with the second term only being 1.19 times the first term.

Table 1: Performance comparison to baselines on classification and regression tasks

| Method | ABCD | | | | HCP | | | | | | UKB | | | | | |
|---|---|---|---|---|---|---|---|---|---|---|---|---|---|---|---|---|
| | Sex | | Intelligence | | Sex | | Age (year) | | Intelligence | | Sex | | Age (year) | | Intelligence | |
| | ACC | AUC | MSE | MAE | ACC | AUC | MSE | MAE | MSE | MAE | ACC | AUC | MSE | MAE | MSE | MAE |
| XGBoost | 69.5 | 76.7 | 0.977 | 0.770 | 68.5 | 75.5 | 14.3 | 3.12 | 0.991 | 0.813 | 79.5 | 87.6 | 48.8 | 5.85 | 1.055 | 0.816 |
| BrainNetCNN[30] | **80.1** | 87.9 | 0.969 | 0.767 | 77.1 | 84.9 | 12.6 | 2.97 | 0.984 | 0.805 | 86.8 | 93.8 | 42.7 | 5.36 | 1.001 | 0.800 |
| VanillaTF[12] | 77.4 | 85.1 | 0.961 | 0.764 | 77.9 | 85.2 | 12.5 | 2.95 | 0.987 | 0.812 | 87.0 | 95.1 | 41.4 | 5.26 | 0.999 | 0.799 |
| BNT[12] | 79.1 | **88.9** | 0.955 | 0.767 | 81.0 | 88.0 | 12.8 | 2.98 | 1.001 | 0.830 | 87.0 | 94.8 | 39.6 | 5.17 | 0.998 | 0.798 |
| TFF[17] | 73.8 | 80.2 | 0.968 | 0.768 | 92.5 | 97.5 | 13.8 | 3.11 | 0.953 | 0.795 | 96.8 | 99.5 | 42.1 | 5.10 | 0.997 | **0.783** |
| SwiFT (ours) | 79.3 | 87.8 | **0.932** | **0.756** | **92.9** | **98.0** | **8.6** | **2.36** | **0.903** | **0.786** | **97.7** | **99.8** | **18.2** | **3.40** | **0.992** | 0.796 |

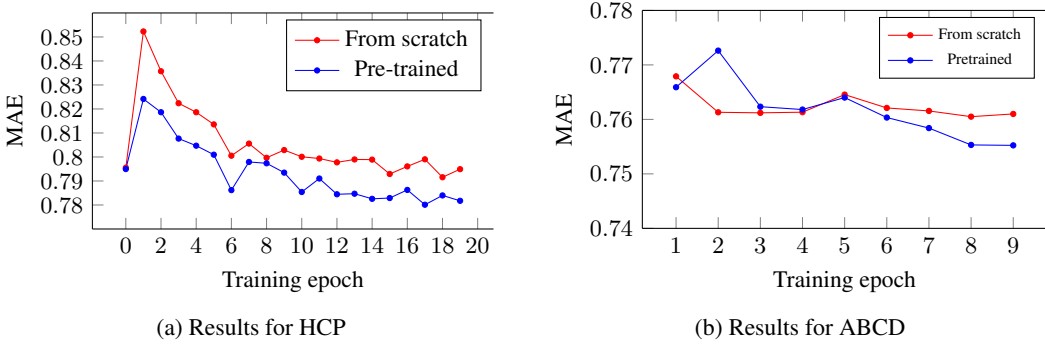

(a) Results for HCP        (b) Results for ABCD

Figure 3: Effect of UKB pre-training on (a) HCP and (b) ABCD intelligence prediction tasks.

Compared to this, with global attention the $2N^2C'$ term becomes 379 times larger than the $12NC'^2$ term, taking up most of the computation budget and creating a bottleneck. For successive stages, $N$ is reduced by a factor of 8, and $C'$ is increased by a factor of 2, resulting in Stage 1 being the most computationally expensive.

### 4.1.2 Baselines

**ROI-based models** We used ROI-based deep learning methods as baseline models, which are listed as BrainNetCNN [30], VanillaTF [12], and Brain Network Transformer (BNT) [12]. These models utilize functional connectivity data as input, which is computed using temporal correlations (Pearson correlation) of every pair of brain regions. To evaluate such methods, we followed the hyper-parameter and implementations of these three models from [12]. In addition, we also employed XGBoost (eXtreme Gradient Boosting) [59] in conjunction with the features described in [13] to compare a traditional machine learning model with that of deep learning-based models. We used the flattened upper triangular correlation matrix as the input for XGBoost.

**TFF (a two-step method)** As mentioned in Section 2, TFF [17] consists of 3D CNNs to reduce the dimensionality of fMRI volumes, which are then passed to a transformer encoder layer for temporal processing. It has been reported that the model achieves state-of-the-art performances in sex classification and age regression in HCP datasets compared to other deep neural networks [11, 60]. However, due to the separate processing of spatial and temporal information, TFF may lose important spatio-temporal information and suffer from performance degradation. Furthermore, since TFF also accepts the fMRI volume as its input, due to memory constraints, we also used the technique of dividing the volume into 20-frame sub-sequences described in Section 4.1.1.

### 4.2 Classification and Regression Results

In Table 1, we compared the performance of SwiFT, which was trained from scratch as in Section 4.1.1, against various baselines mentioned above on sex classification and age, intelligence regression tasks. On the sex classification task, SwiFT outperforms all of the baselines on the HCP and UKB datasets while showing competitive results against the best ROI-based models on the ABCD dataset. On the regression tasks, we observe SwiFT outperforms all baselines significantly for the age prediction tasks, although all models still have a large room for improvement on the UKB intelligence prediction task. We believe our quantitative results clearly underscore the effectiveness of the end-to-end training of our SwiFT model.

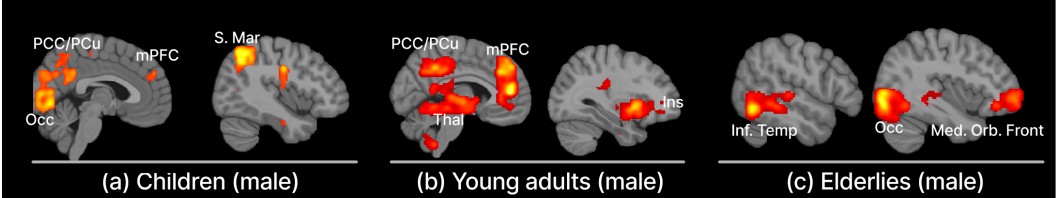

Figure 4: Interpretation maps with Integrated Gradients (IG) for sex classification. (Sagittal plane) **(a)** ABCD **(b)** HCP **(c)** UKB

### 4.3 Effects of Pre-training on Downstream Tasks

To demonstrate the effectiveness of contrastive pre-training described in Section 3.2, SwiFT pre-trained on a larger dataset (UKB) was fine-tuned on a smaller dataset (HCP), and a comparable-sized dataset (ABCD) for the intelligence prediction task, which has room for improvement compared to sex and age prediction tasks. The model was pre-trained using the combination of the instance contrastive loss function ($\mathcal{L}_{IC}$) and the local-local temporal contrastive loss function ($\mathcal{L}_{LL}$) such that the training objective is to minimize the sum ($\mathcal{L}_{IC} + \mathcal{L}_{LL}$). The feature representations used in the loss calculation were obtained in the same manner as other tasks; by applying a global average pooling layer on the output feature map of Stage 4, followed by an MLP head. Figure 3 depicts the model's performance for each training epoch during the fine-tuning process. The performances are averaged over three splits. The results from a model trained from scratch (Section 4.2) are also shown for comparison. In HCP, the pre-trained model consistently performs better during the early stage of the training compared to the model trained from scratch. In ABCD, on the other hand, the pre-trained model exhibits dramatic drops in performance at the early stage of training and gradually attains a better performance at the later stage of fine-tuning. This initial worse performance might result from the sub-optimal training hyper-parameters for fine-tuning, such as the sub-optimal initial learning rate. Overall, we believe our results suggest that the contrastive pre-training of SwiFT on a larger dataset has a promising effect on improving downstream performance, and we plan to investigate more extensively on this topic in our future research.

### 4.4 Interpretation Results

Using an Integrated Gradient with Smoothgrad sQuare (IG-SQ) implemented in Captum framework [15, 61, 62], we identified the brain regions showing high explanatory power on the sex classification task. We acquired 4D IG-SQ maps from test sets and filtered out incorrectly predicted samples. To find the spatial patterns of the brain showing explanatory power across subjects, we normalized the IG-SQ maps, smoothing the maps with a Gaussian filter. Then, we averaged the maps over time dimensions and across subjects.

Figure 4 shows the brain regions contributing to successful sex classification. These include, in children (ABCD), the medial prefrontal gyrus (mPFC), posterior cingulate cortex (PCC), precuneus (PCu), and parietal gyrus (the default mode network). In young adults (HCP), similar brain regions were observed with a broader and higher intensity in mPFC, while showing unique brain regions such as the thalamus and insular cortex. In middle and old-aged adults (UKB), we acquired the highest IG values in the inferior temporal gyrus and medial orbitofrontal cortex. Of note, the results confirm those regions implicated in prior brain sex difference literature [63, 64, 65, 66].

### 4.5 Model Efficiency

Table 2: Efficiency of 4D fMRI Transformers

| Method | # Param. | FLOPs | Throughput (samples/sec) |
|--------|----------|-------|--------------------------|
| TFF    | 729M     | 40.72G | 53.60 |
| SwiFT  | 4.64M    | 2.62G  | 104.46 |

In Table 2, we compared the computational efficiency of SwiFT against TFF by using dummy data with random numbers. Each sample consists of 20 volumes with the shape of $96 \times 96 \times 96$.

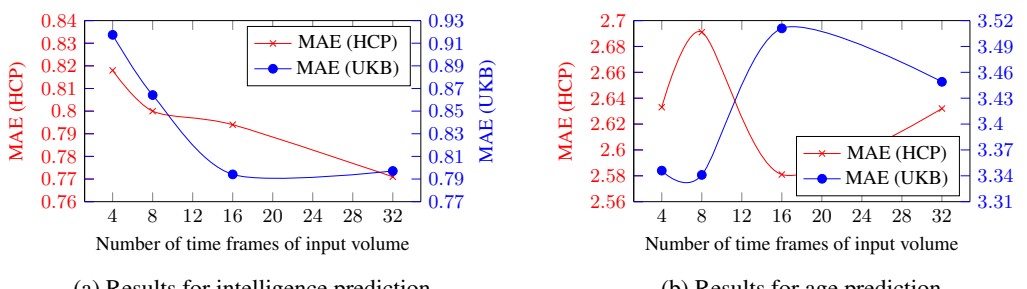

(a) Results for intelligence prediction.

(b) Results for age prediction.

Figure 5: Effect of the number of time frames of the input fMRI volume on (a) intelligence prediction and (b) age prediction tasks.

Floating point operations (FLOPs) were used to estimate the amounts of multiply-add computations required for processing the 4D fMRI volumes. The throughput (samples/s) denotes the number of 20-volume samples processed per second, calculated using a single NVIDIA A100 GPU. For an accurate measurement, the throughput was measured using synchronized timing with an initial GPU warmup step and was repeated 100 times. The results show that SwiFT has 158.4 times fewer parameters, requires 15.5 times fewer multi-add operations, and processes input data 1.94 times faster than TFF. This result shows SwiFT is much more computationally efficient compared to TFF while also attaining better performance as given in Table 1.

### 4.6  Effect of Input Time Sequence Length

To justify our use of a 4D model and investigate the consequences of dividing the input fMRI volume into sub-sequences, we evaluated the effect of the input fMRI volume's sequence length (number of time frames). To ensure a fair comparison, we kept the model architecture and hyper-parameters, such as the local window size, constant. In Figure 5, we compared the model performance by input sequence lengths (4, 8, 16, 32 time frames) on intelligence and age prediction tasks, respectively. In the intelligence prediction task — which is a challenging task considering the MAE of about 0.8 (z-score) was only slightly better than the variance of one — longer sequence lengths (16 and 32) led to better results in both young adults (HCP) and elders (UKB). Also, in age prediction, for young adults, we found the performance peaking at 16 time frames. However, in predicting the age of elders, longer sequence lengths (16, 32) resulted in poorer performance in age prediction. Namely, the findings of the former three cases showed that the longer sequence lengths enabled better learning of temporal dynamics needed to predict intelligence in young adults and elders and age in young adults. However, the last case of the age prediction task in elders showed that the benefit might not be generalizable to all the cases.

These findings point out that the optimal value of the input time sequence length may vary depending on the given task and dataset, and our choice to employ constant-sized sub-sequences may have proven beneficial in light of this variability. A more detailed analysis can be found in Appendix C.2.

## 5  Concluding Remarks

Investigating the spatiotemporal dynamics of the human brain poses a formidable challenge owing to the lack of powerful analytics permitting individual-level prediction of cognitive or clinical outcomes such as psychiatric or neurological diseases. In this study, we present SwiFT, an efficient Transformer model designed for high-dimensional 4D brain functional MRI, aimed at learning these spatiotemporal dynamics and predicting biological and cognitive outcomes. Throughout various tasks, our method consistently outperforms the state-of-the-art ROI-based and two-step based methods in the domain. Furthermore, compared to TFF, a Transformer-based baseline that takes raw fMRI as input, SwiFT uses significantly less memory and training time. Lastly, our IG-SQ interpretation results show the feasibility of interpreting the spatial patterns of the functional representations from SwiFT that contribute to a given task. We believe the simple and effective end-to-end learning capability of our SwiFT has the potential to significantly contribute to enhancing the scalability of spatiotemporal learning of fMRI in both computational and clinical neuroscience research. For future work, we plan to more extensively test the effectiveness of pre-training SwiFT on large-scale datasets.

# 6 Acknowledgements

This work was supported by the U.S. Department of Energy (DOE), Office of Science (SC), Advanced Scientific Computing Research program under award DE-SC-0012704 and used resources of the National Energy Research Scientific Computing Center, a DOE Office of Science User Facility supported by the Office of Science of the U.S. Department of Energy under Contract No. DE-AC02-05CH11231 using NERSC award ASCR-ERCAP0023081. Also, this research used resources of the Argonne Leadership Computing Facility, which is a U.S. Department of Energy Office of Science User Facility supported under Contract DE-AC02-06CH11357.

This work was also supported in part by the National Research Foundation of Korea (NRF) grant funded by the Korean government (MSIT) [No. 2021R1C1C1006503, 2021K1A3A1A2103751212, 2021M3E5D2A01022515, RS-2023-00265406, 2021M3E5D2A01024795, 2021R1A2C2007884], by Creative-Pioneering Researchers Program through Seoul National University (No. 200-20230058), and by Institute of Information & communications Technology Planning & Evaluation (IITP) grant funded by the Korea government (MSIT) [No.2021-0-01343, No.2021-0-02068, No.2022-0-00113, No.2022-0-00959]. Taesup Moon is also supported in part by ASRI / INMC at Seoul National University.

This research has been conducted using data from UK Biobank, a major biomedical database (https://www.ukbiobank.ac.uk). Our project ID of the UK Biobank is 32575. Data used in the preparation of this article were obtained from the Adolescent Brain Cognitive Development (ABCD) Study (https://abcdstudy.org), held in the NIMH Data Archive (NDA). This is a multisite, longitudinal study designed to recruit more than 10,000 children aged 9-10 and follow them over 10 years into early adulthood. The ABCD Study® is supported by the National Institutes of Health and additional federal partners under award numbers U01DA041048, U01DA050989, U01DA051016, U01DA041022, U01DA051018, U01DA051037, U01DA050987, U01DA041174, U01DA041106, U01DA041117, U01DA041028, U01DA041134, U01DA050988, U01DA051039, U01DA041156, U01DA041025, U01DA041120, U01DA051038, U01DA041148, U01DA041093, U01DA041089, U24DA041123, U24DA041147. A full list of supporters is available at (https://abcdstudy.org/federal-partners.html). A listing of participating sites and a complete listing of the study investigators can be found at (https://abcdstudy.org/consortium_members/). ABCD consortium investigators designed and implemented the study and/or provided data but did not necessarily participate in the analysis or writing of this report. This manuscript reflects the views of the authors and may not reflect the opinions or views of the NIH or ABCD consortium investigators. The ABCD data repository grows and changes over time. The ABCD data used in this report came from http://dx.doi.org/10.15154/1503209. DOIs can be found at https://nda.nih.gov/abcd/abcd-annual-releases.html. Data were provided [in part] by the Human Connectome Project, WU-Minn Consortium (Principal Investigators: David Van Essen and Kamil Ugurbil; 1U54MH091657) funded by the 16 NIH Institutes and Centers that support the NIH Blueprint for Neuroscience Research; and by the McDonnell Center for Systems Neuroscience at Washington University.

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

# A Performance Comparison with Standard Deviation

Here, we detail the standard deviation of the results posted in Table 1 (manuscript), which compares the performance of SwiFT against other baseline models on various downstream tasks. All of the experiments were performed using three pre-determined random data splits for each dataset, which was shared across all models for a fair comparison. The following Tables (A.1, A.2, A.3) show the performance posted in Table 1 (manuscript) along with the standard deviation among the three splits. From the tables, we can conclude that SwiFT outperforms all baseline models above the margin of variability at the following tasks: ABCD intelligence, HCP sex, HCP age, HCP intelligence, UKB sex, and UKB age prediction tasks. Note that the age prediction task was not carried out on the ABCD dataset due to the narrow age range of 9 to 11 years, making it hard to obtain meaningful results from the experiments.

Table A.1: Performance of various models on the ABCD dataset with standard deviation

| Method | Dataset: ABCD | | | |
| | Sex | | Intelligence | |
| | ACC | AUC | MSE | MAE |
| --- | --- | --- | --- | --- |
| XGBoost | $69.5_{\pm 0.59}$ | $76.7_{\pm 0.86}$ | $0.977_{\pm 0.037}$ | $0.770_{\pm 0.016}$ |
| BrainNetCNN [30] | $\mathbf{80.1}_{\pm 0.69}$ | $87.9_{\pm 0.37}$ | $0.969_{\pm 0.042}$ | $0.767_{\pm 0.016}$ |
| VanillaTF [12] | $77.4_{\pm 2.47}$ | $85.1_{\pm 3.21}$ | $0.961_{\pm 0.050}$ | $0.764_{\pm 0.025}$ |
| BNT [12] | $79.1_{\pm 0.80}$ | $\mathbf{88.9}_{\pm 0.64}$ | $0.955_{\pm 0.058}$ | $0.767_{\pm 0.025}$ |
| TFF [17] | $73.8_{\pm 1.13}$ | $80.2_{\pm 1.06}$ | $0.968_{\pm 0.024}$ | $0.768_{\pm 0.009}$ |
| SwiFT (ours) | $79.3_{\pm 1.29}$ | $87.8_{\pm 1.31}$ | $\mathbf{0.932}_{\pm 0.017}$ | $\mathbf{0.756}_{\pm 0.009}$ |

Table A.2: Performance of various models on the HCP dataset with standard deviation

| Method | Dataset: HCP | | | | | |
| | Sex | | Age | | Intelligence | |
| | ACC | AUC | MSE | MAE | MSE | MAE |
| --- | --- | --- | --- | --- | --- | --- |
| XGBoost | $68.5_{\pm 3.03}$ | $75.5_{\pm 3.14}$ | $14.3_{\pm 1.61}$ | $3.12_{\pm 0.165}$ | $0.991_{\pm 0.084}$ | $0.813_{\pm 0.032}$ |
| BrainNetCNN [30] | $77.1_{\pm 2.33}$ | $84.9_{\pm 2.28}$ | $12.6_{\pm 0.74}$ | $2.97_{\pm 0.153}$ | $0.984_{\pm 0.034}$ | $0.805_{\pm 0.016}$ |
| VanillaTF [12] | $77.9_{\pm 2.08}$ | $85.2_{\pm 0.89}$ | $12.5_{\pm 1.15}$ | $2.95_{\pm 0.182}$ | $0.987_{\pm 0.039}$ | $0.812_{\pm 0.014}$ |
| BNT [12] | $81.0_{\pm 3.11}$ | $88.0_{\pm 3.10}$ | $12.8_{\pm 0.89}$ | $2.98_{\pm 0.155}$ | $1.001_{\pm 0.009}$ | $0.830_{\pm 0.014}$ |
| TFF [17] | $92.5_{\pm 1.12}$ | $97.5_{\pm 1.77}$ | $13.8_{\pm 1.58}$ | $3.11_{\pm 0.200}$ | $0.953_{\pm 0.074}$ | $0.795_{\pm 0.028}$ |
| SwiFT (ours) | $\mathbf{92.9}_{\pm 1.51}$ | $\mathbf{98.0}_{\pm 1.79}$ | $\mathbf{8.6}_{\pm 0.57}$ | $\mathbf{2.36}_{\pm 0.114}$ | $\mathbf{0.903}_{\pm 0.077}$ | $\mathbf{0.786}_{\pm 0.030}$ |

Table A.3: Performance of various models on the UKB dataset with standard deviation

| Method | Dataset: UKB | | | | | |
| | Sex | | Age | | Intelligence | |
| | ACC | AUC | MSE | MAE | MSE | MAE |
| --- | --- | --- | --- | --- | --- | --- |
| XGBoost | $79.5_{\pm 1.28}$ | $87.6_{\pm 0.94}$ | $48.8_{\pm 1.01}$ | $5.85_{\pm 0.046}$ | $1.055_{\pm 0.199}$ | $0.816_{\pm 0.078}$ |
| BrainNetCNN [30] | $86.8_{\pm 0.19}$ | $93.8_{\pm 0.31}$ | $42.7_{\pm 0.17}$ | $5.36_{\pm 0.113}$ | $1.001_{\pm 0.141}$ | $0.800_{\pm 0.060}$ |
| VanillaTF [12] | $87.0_{\pm 1.31}$ | $95.1_{\pm 0.37}$ | $41.4_{\pm 1.16}$ | $5.26_{\pm 0.142}$ | $0.999_{\pm 0.144}$ | $0.799_{\pm 0.059}$ |
| BNT [12] | $87.0_{\pm 1.10}$ | $94.8_{\pm 0.46}$ | $39.6_{\pm 1.07}$ | $5.17_{\pm 0.092}$ | $0.998_{\pm 0.139}$ | $0.798_{\pm 0.058}$ |
| TFF [17] | $96.8_{\pm 0.25}$ | $99.5_{\pm 0.06}$ | $42.1_{\pm 4.80}$ | $5.10_{\pm 0.331}$ | $0.997_{\pm 0.123}$ | $\mathbf{0.783}_{\pm 0.046}$ |
| SwiFT (ours) | $\mathbf{97.7}_{\pm 0.31}$ | $\mathbf{99.8}_{\pm 0.04}$ | $\mathbf{18.2}_{\pm 0.94}$ | $\mathbf{3.40}_{\pm 0.083}$ | $\mathbf{0.992}_{\pm 0.105}$ | $0.796_{\pm 0.044}$ |

# B Implementation Details

**SwiFT** To obtain the results detailed in Section 4.2 (manuscript), we trained SwiFT from scratch using the following configuration:

- *Optimizer*: AdamW using a cosine decay learning rate scheduler with a linear warm-up (around 5% of total iterations)
- *Learning rate*: After the warm-up, an initial learning rate of $10^{-5}$ for classification tasks and $5 \times 10^{-5}$ for regression tasks on the HCP dataset and $10^{-5}$ for regression tasks on the ABCD dataset.
- *Mini-batch size*: Mini-batch of size 8

- *Epochs*: 10 epochs of training for the sex classification and intelligence prediction tasks, and a maximum of 30 epochs for the age prediction task.

After training, we selected the model instances with the highest validation AUC or lowest validation MSE to report the performance on the test dataset. The model was trained using four NVIDIA A100 GPUs using the distributed data-parallel (DDP) strategy provided by Pytorch Lightning, with a single training session typically lasting from 4 to 30 hours depending on the dataset and whether data augmentation was used during training. The model can be trained on an NVIDIA A5000 GPU with 24GB memory without a problem.

To obtain the results detailed in Section 4.3. (manuscript), the model was pre-trained using the following:

- *Optimizer*: AdamW optimizer using a cosine decay learning rate scheduler with 2000 steps of linear warm-up
- *Learning rate*: After warm-up, an initial learning rate of $10^{-5}$
- *Mini-batch size*: 3
- *Epochs*: six epochs of training

After pre-training, we fine-tuned the model without a learning rate scheduler, using 1/10 of the initial learning rate used for from-scratch training.

**TFF**    We used the same data splits to compare our baseline 4D Transformer, TFF, with SwiFT. We followed the number of attention heads (16) and embedding size (2,640) proposed by [17]. To alleviate over-fitting, we applied data augmentation methods to the brain images, such as Gaussian blur and additive Gaussian noise implemented by imgaug[67]. Since TFF requires more computational resources than SwiFT to run the codes, at least 8 hours of training were required using 2 nodes with 4 A100 GPUs.

We trained TFF with the following training setup:

- *Optimizer*: Adam optimizer using a cosine decay learning rate scheduler with a linear warm-up by 5% of total iterations
- *Learning rate*: After warm-up, an initial learning rate of $10^{-4}$
- *Mini-batch size*: 32
- *Epochs*: 10 epochs of training

**ROI-based models**    The four ROI-based model baselines, XGBoost [59], BrainNetCNN [30], VanillaTF [12], and Brain Network Transformer [12] were reproduced for our experiments. The reproductions were based on the official code of [12]. However, since the preprocessing codes for the ABCD dataset were not provided by [12], we followed the preprocessing steps described in [12], potentially causing some differences. This obscurity in the preprocessing step is suspected to be one of the reasons for the slight performance gap of the BNT model between our experiments and the results posted in the original paper [12], despite utilizing the same ABCD dataset.

We trained BrainNetCNN, VanillaTF, and Brain Network Transformer models with the following setup:

- *Optimizer*: Adam optimizer using a cosine decay learning rate scheduler
- *Learning rate*: Learning rate of $5 \times 10^{-5}$
- *Mini-batch size*: Mini-batch of size 16
- *Epochs*: 200 epochs of training

We used grid search for hyper-parameter tuning of XGBoost, adjusting the maximum depth and minimal child weight, gamma, learning rate, and colsample by tree. In addition, we conducted 5-fold cross-validation. Hyperparameters are tuned with the following setup:

- *Maximum depth*: Chosen between 3 and 6
- *Minimal child weight*: Chosen between 1 and 7
- *Gamma*: Chosen between 0.0 and 0.4
- *Learning rate*: Chosen between 0.05 and 0.3
- *Colsample by tree*: Chosen between 0.6 and 0.9

**Software Version**    The major software used for our experiments are as the following:

Table A.4: Performance comparison of SwiFT for different positional embedding methods.

| Dataset | Method | Sex | | Age | | Intelligence | |
|---|---|---|---|---|---|---|---|
| | | ACC | AUC | MSE | MAE | MSE | MAE |
| HCP | Relative | $89.8_{+1.87}$ | $95.9_{+1.21}$ | $9.0_{+0.56}$ | $2.44_{+0.116}$ | $0.908_{+0.009}$ | $0.775_{+0.011}$ |
| | Absolute | $\mathbf{92.9}_{+1.51}$ | $\mathbf{98.0}_{+1.79}$ | $\mathbf{8.6}_{+0.57}$ | $\mathbf{2.36}_{+0.114}$ | $\mathbf{0.903}_{+0.077}$ | $\mathbf{0.786}_{+0.030}$ |
| ABCD | Relative | $\mathbf{80.2}_{+1.65}$ | $\mathbf{88.9}_{+0.26}$ | N/A | | $0.936_{+0.029}$ | $0.761_{+0.013}$ |
| | Absolute | $79.3_{+1.29}$ | $87.8_{+1.31}$ | | | $\mathbf{0.932}_{+0.017}$ | $\mathbf{0.756}_{+0.009}$ |
| UKB | Relative | $97.5_{+0.10}$ | $99.8_{+0.05}$ | $19.4_{+0.53}$ | $3.53_{+0.071}$ | $1.019_{+0.083}$ | $0.807_{+0.035}$ |
| | Absolute | $\mathbf{97.7}_{+0.31}$ | $99.8_{+0.04}$ | $\mathbf{18.2}_{+0.94}$ | $\mathbf{3.40}_{+0.083}$ | $\mathbf{0.992}_{+0.105}$ | $\mathbf{0.796}_{+0.044}$ |

Table A.5: Efficiency comparison of SwiFT for different positional embedding methods.

| Method | # Param. | FLOPs | Throughput |
|---|---|---|---|
| Relative | 4.66M | 2.62G | 94.17 |
| Absolute | 4.64M | 2.62G | 104.16 |

- python 3.10.4
- pytorch 1.12.1
- pytorch-lightning 1.6.5
- monai 1.1.0
- neptune-client 0.16.4
- scipy 1.8.1
- torchvision 0.13.1
- torchaudio 0.12.1

The full dependency list is released alongside the code.

# C   Additional Experiments

## C.1   Comparison of Positional Embedding Methods

We discuss the impact of the switch from a relative positional bias scheme used in most Swin Transformer variants [19, 23, 20] to an absolute positional embedding scheme, as detailed in the paragraph "4D absolute positional embedding" of Section 3.1 (manuscript). We implemented the 4D relative positional bias scheme by extending the 3D relative positional bias described in [23]. Given a window with dimensions of $P \times M \times M \times M$, the 4D relative positional bias $B \in \mathbb{R}^{P^2 \times M^2 \times M^2 \times M^2}$ for each self-attention head is integrated as

$$\text{Attention}(Q, K, V) = \text{SoftMax}(QK^T/\sqrt{d} + B)V, \tag{1}$$

where $Q, K, V \in \mathbb{R}^{PM^3 \times C'}$ are the query, key, value matrices and $C'$ is the channel number. A parameterized bias matrix $\hat{B} \in \mathbb{R}^{(2P-1) \times (2M-1) \times (2M-1) \times (2M-1)}$ is used to calculate the values in $B$ since there are only $2P - 1$ or $2M - 1$ possible position differences for every axis. A separate parameterized bias matrix is used for each attention head at each layer of the Transformer.

Table A.4 shows the overall performance comparison of SwiFT between the absolute positional embedding scheme and the relative positional bias scheme for the HCP, ABCD, and UKB datasets. Comparing the performance of both methods, we found that the absolute positional embedding scheme shows a better performance in most cases. Note that the age prediction task on the ABCD dataset was not tested due to the same reasons detailed in Section A.

Additionally, we compared the efficiency of the two methods in Table A.5 through the number of parameters, the number of FLOPs per forward pass, and the throughput. Throughput measures how many fMRI sub-sequences of length 20 the model processes per second during inference on a single A100 GPU. The absolute positional embedding scheme is more memory efficient as it only requires parameters at the beginning of each stage. In addition, the absolute positional embedding scheme is also more computationally efficient as it does not require the 4D relative positional bias $B$ to be reconstructed during each self-attention computation, resulting in a 9.6% throughput improvement. Overall, we conclude that the absolute positional embedding scheme is appropriate for our tasks compared to the relative positional bias scheme.

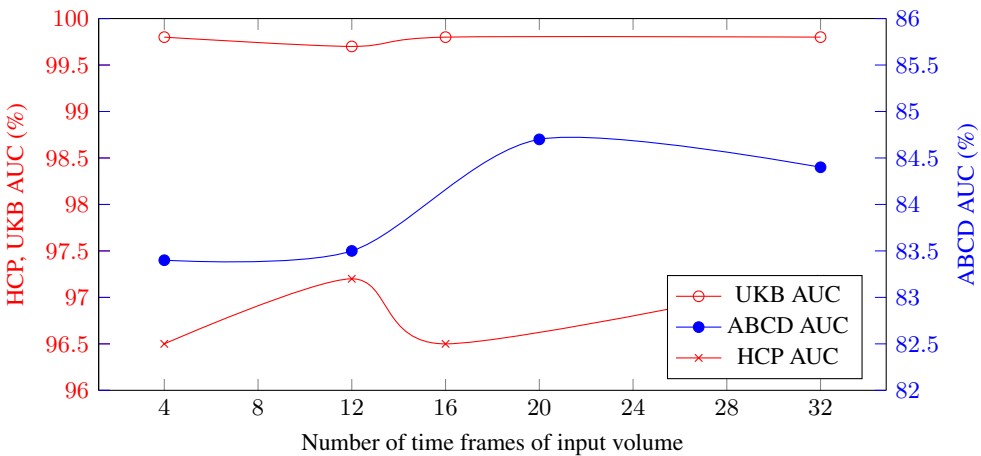

Figure A.1: Effect of the number of time frames of the input fMRI volume on sex classification.

## C.2   Effect of Input Time Sequence Length for Sex Classification

In our setup, instead of inputting the entire fMRI sequence of a subject all at once, we have opted to divide it into 20-frame sub-sequences and use the divided sub-sequences as the input to our model. This was mainly due to memory constraints and to standardize the number of input tokens to our model. We have investigated the effect of changing the length of this input sub-sequence in section 4.6. Here, we present our experiments on the sex classification task (Figure A.1) and elaborate as to why the optimal length seems to change for each dataset and task.

The SwiFT model was trained with a varying number of input fMRI volumes, with 20 volumes being the default configuration used for the previous experiments. We adjusted the mini-batch size to keep the total number of training iterations per epoch constant. All the performances in the figure are averaged performances from three pre-determined data splits. We note that the training data augmentation on the ABCD dataset was not applied in this experiment to keep the training environment consistent compared to other datasets, and thus the model has a lower performance compared to the results posted in Section 4.2 (manuscript).

The performance of the sex classification task on the HCP and UKB dataset is already saturated near 100% AUC, and thus we observed that changing the number of input time frames has a small or negligible effect on the performance, ranging within one standard deviation. However, for the ABCD dataset, the performance of the model has a larger fluctuation. The performance peaked at around 20 input frames, which is the default number of input volumes used for the other experiments.

Inputting a longer time sequence into the model would allow the model to attend to longer sequences at once, at the expense of bloating the model with more parameters and causing it to overfit. Therefore in a setting where we are dealing with a limited number of training data, using a shorter sub-sequence has a possibility to be beneficial to the model's generalization capabilities. We also note that the total amount of information used for our predictions remains the same regardless of the input time sequence length, as the averaged logits from all sub-sequences of the subject are used for the prediction. Additionally, the datasets used for this work are resting-state fMRI scans, where the image does not change drastically over time. In the future, it would be interesting to investigate the effect of input time sequence length on more temporally dynamic datasets, such as task-based fMRI scans.

## C.3   Time-window Analysis

In accordance with the details provided in Section 4.1.1, the SwiFT model was trained by processing individual 20-frame time windows (sub-sequences) of fMRI data sequentially, while sliding the time windows to encompass the entire fMRI dataset. For inference purposes, the predictions obtained from the fMRI time windows were aggregated by averaging the logit values of each subject. This averaged value was then utilized as the final prediction for the respective subject. To ensure that the predictions

of time windows from the same subjects are homogeneous and the final predictions are not decided by a few noisy time windows, we verified how many subjects exhibited distinct predictions among their time windows using the sex classification task of ABCD, HCP, and UKB datasets. Each subject of ABCD, HCP, and UKB has 18, 60, and 24 time windows.

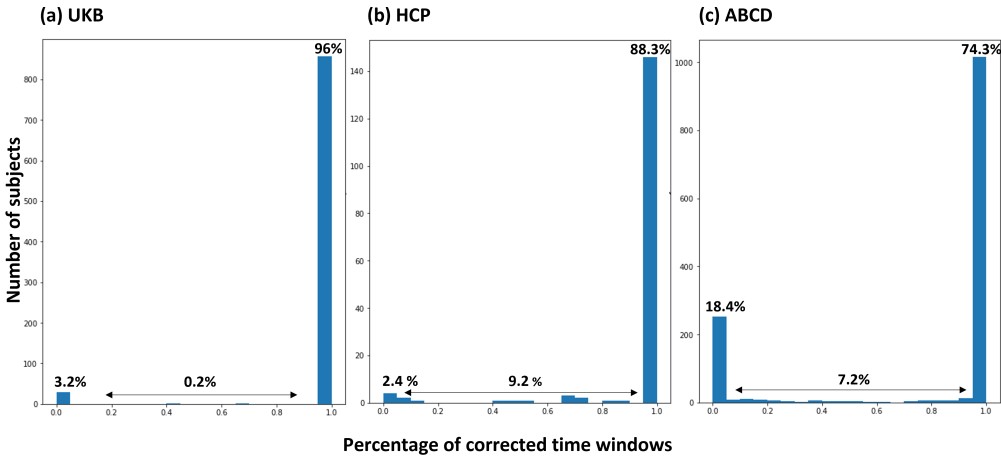

Figure A.2: Inner-subject accuracy of sex classification.

As seen in the histogram in Figure A.2, over 90% of subjects showed identical predictions among the time windows (0.992 for UKB, 0.907 for HCP, and 0.927 for ABCD). Of note, 1.0 on the x-axis denotes that the predictions from the time windows of the subjects are perfectly correct, while 0.0 means none of the time windows exhibited correct predictions. This suggests that the predictions of time windows are homogeneous and that the final predictions of each subject are not biased toward a few time windows.

## D   SwiFT's Ability to Capture Long-range Interactions

Owing to the local nature of the attention mechanism, there could be concerns about SwiFT not being able to capture long-range interactions between spatially/temporally separated tokens. However, there are some built-in mechanisms that could learn about long-range communications between tokens.

On the spatial dimensions, the Swin Transformer model can cover large ranges on the later parts of its layers through the patch merging step. On our configuration, after the initial patch embedding step where $6 \times 6 \times 6 = 216$ neighboring voxels are embedded into a patch, a $4 \times 4 \times 4$ window can cover a range equivalent to $24 \times 24 \times 24$ voxels from our $96 \times 96 \times 96$ input. However, after the patch merging step where 8 neighboring tokens are merged, the same window can cover a range equivalent to $48 \times 48 \times 48$ voxels. After another patch merging step, on stage 3, this grows to $96 \times 96 \times 96$ which is already the same size as our entire input. From this point on, a single window covers the entire span of our input in the spatial dimension, thanks to the patch merging step.

On the temporal dimension, where patch merging does not occur on the current version of the model, we needed another method that can ensure long-range interaction between tokens. The shifting window allows a token to interact with tokens that are about 10 time frames away from our setup, but to allow longer range interactions we made the model do a full global attention on the final stage as mentioned in Section 3.1 (manuscript). Thus, the model processes the entire temporal range in the final stage, allowing it to introduce long-range temporal interactions in the end. We acknowledge that there could be more effective methods to allow long-range interactions, and it would be an exciting problem to tackle when we are dealing with more temporally dynamic datasets.

# E   Loss Curves

Here we detail the model's performance on the validation set during the training procedure in order to provide a reference for our training procedure. Note that the validation MSE spikes for ABCD intelligence prediction are most likely due to the restarts of the learning rate scheduler.

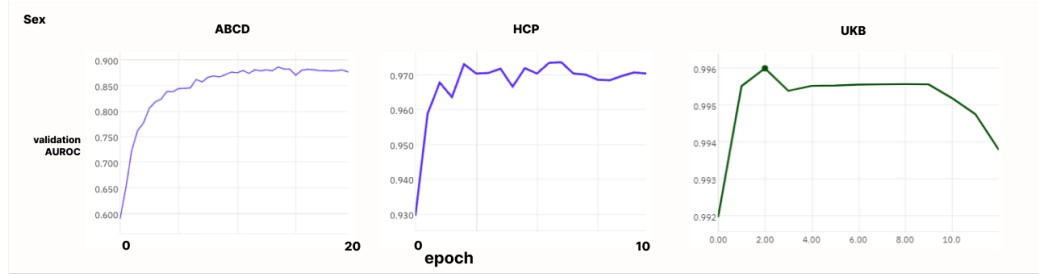

(a) Validation AUROC per training epoch for sex classification

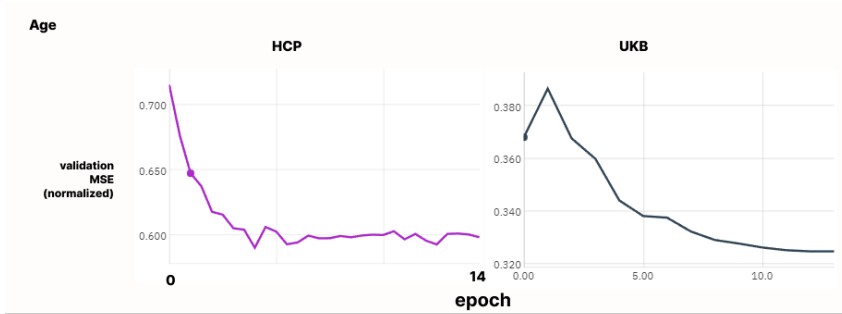

(b) Validation MSE per training epoch for age prediction

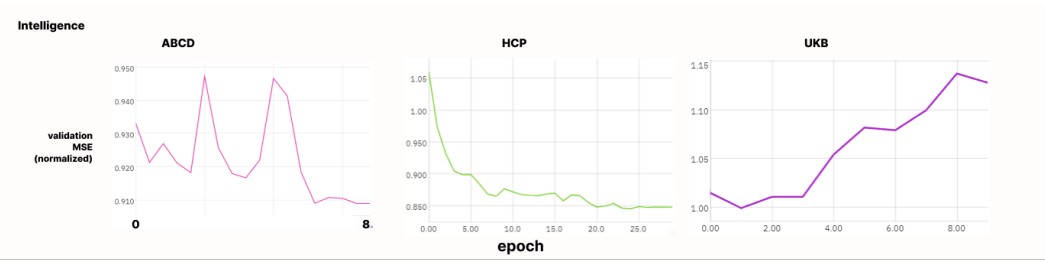

(c) Validation MSE per training epoch for intelligence prediction

Figure A.3: Performance on the validation set during the training procedure.

# F   Limitations

The biggest limitation of our research is that the advantages of self-supervised pre-training were not evident in some tasks. The pre-training was found to foster early convergence during fine-tuning on some downstream tasks, such as HCP and ABCD intelligence prediction, but in some other cases, the performance gain from pre-training was not substantial. This result can be attributed to the distinct age range between UK Biobank used for pre-training and the other two datasets for downstream tasks, ABCD and HCP. In addition, the scanner effect of fMRI, which confounds the biological and cognitive features, can also hamper knowledge transfer. Otherwise, the performance increase might be limited because we have already reached the upper bounds for the sex and age prediction tasks. Previously, few studies targeted a transfer learning of 4D fMRI between different data sources. We suggest that various sources of the dataset should be included during pre-training in supervised or unsupervised ways. Though the observed performance increase from transfer learning was limited, we established the foundation for large-scaling pre-training for 4D fMRI. Consequently, optimizing

self-supervised pre-training methodology for fMRI with larger amounts of datasets are promising future research topic.

## G  Licences

We used existing assets as follows:

- TFF [17]: https://github.com/GonyRosenman/TFF, No explicit license.
- Brain Network Transformer [12] : https://github.com/Wayfear/BrainNetworkTransformer, MIT License.
- Swin Transformer [19] : https://github.com/microsoft/Swin-Transformer, MIT License.
- MonAI : https://github.com/Project-MONAI/MONAI, Apache License.
- TCLR [40]: https://github.com/DAVEISHAN/TCLR, No explicit license

