# OpenReview forum: "SwiFT: Swin 4D fMRI Transformer"
_NeurIPS.cc/2023/Conference — NeurIPS 2023 poster_

### Official Review · Reviewer_st1h · 2023-07-02

**Soundness:** 3 good
**Presentation:** 3 good
**Contribution:** 3 good
**Rating:** 6
**Confidence:** 4

**Summary:**

This paper extends Swin Transformer to 4D brain functional MRI data. Unlike existing deep learning methods in the neuroscience field that either use an ROI-based method (taking as inputs functional connectivity) or a two-step method (3D spatial encoding followed by temporal encoding), the proposed model SwiFT takes as inputs 4D fMRI data and can be trained in an end-to-end manner. This also allows pre-training using contrastive losses. Experiments on multiple large human functional brain imaging datasets indicate the effectiveness of SwiFT compared to prior ROI-based and two-stage methods.

**Strengths:**

1. Originality: this paper is the first attempt at extending Swin Transformer to 4D fMRI data.
2. Methods are technically sound. Experiments are carefully designed to support the authors’ claims.
3. While the building blocks of SwiFT exist in the literature, I consider this paper a significant contribution because 1) it extends Swin Transformer to 4D fMRI data and 2) it addresses a technically challenging task of modeling fMRI data in an end-to-end manner.

**Weaknesses:**

Due to memory constraint, the fMRI sequence needs to be divided into shorter sub-sequences and the model predictions for sub-sequences are aggregated through averaging. In my opinion, this is a limitation of the method and should be discussed.

**Questions:**

1. In Section 3.2 “Instance contrastive loss”, what’s a clip? Please define it.
2. In page 6 lines 230-231, the authors mention that the average performance across 3 splits are reported. But in Appendix page 2 lines 26-27, the authors state that they report the performance on the test set. Please clarify. Also, is the train-validation-test data split patient-wise (i.e., different splits have different patients)? If not, the tasks are much easier and the results may be over-saturated.
3. Ablation studies: What’s the effect of the contrastive losses? Also, the ablation of absolute vs relative position biases is hidden in the Appendix, which should be mentioned in section 3.1 “4D absolute positional embedding”.

**Limitations:**

Limited performance grain from pre-training is discussed in Appendix.

---

> ### Author Rebuttal · Authors · 2023-08-10
>
> Thank you for the review and giving us some helpful pointers. Here is our response to the questions and concerns:
>
> >*Due to memory constraint, the fMRI sequence needs to be divided into shorter sub-sequences and the model predictions for sub-sequences are aggregated through averaging. In my opinion, this is a limitation of the method and should be discussed.*
>
> - We understand your concerns regarding the issue, however we believe that there to be a trade-off for using sub-sequences. Please refer to the general rebuttal regarding the matter.
>
> > *In Section 3.2 “Instance contrastive loss”, what’s a clip? Please define it.*
>
> - We apologize for not stating it clearly. A *clip* refers to an fMRI sub-sequence that acts as an input for our model. Our goal for the contrastive pre-training process is to train the model to differentiate clips (inputs) that come from different subjects or different time frames. We would be happy to clarify the term for the final version of our paper.
>
> > *In page 6 lines 230-231, the authors mention that the average performance across 3 splits are reported. But in Appendix page 2 lines 26-27, the authors state that they report the performance on the test set. Please clarify. Also, is the train-validation-test data split patient-wise (i.e., different splits have different patients)? If not, the tasks are much easier and the results may be over-saturated.*
> - Thank you for the suggestion. To clarify our setup, we have created 3 randomly generated splits that separate our subjects to training, validation, and test sets with a 70:15:15 ratio. For example, if we had 1000 subjects in a dataset, for each split, 700 of them would be used for training, 150 would be used for validation, and 150 would be used for testing. For each split separately, the model was trained on subjects from the training set and after each epoch evaluated on the validation set, the best model iteration was chosen based on the performance on the validation set, and the chosen iteration was evaluated on the test set. The evaluation results (on the test set) from the 3 splits were averaged and reported on our paper.
>
> >*Ablation studies: What’s the effect of the contrastive losses? Also, the ablation of absolute vs relative position biases is hidden in the Appendix, which should be mentioned in section 3.1 “4D absolute positional embedding”.*
>
> - Regarding the effect of the contrastive losses, we have conducted experiments comparing a SwiFT model trained from scratch and a contrastive pre-trained SwiFT model in section 4.3 and appendix C.3.
> - Thank you for the suggestion and we would mention the positional embedding experiment in section 3.1 of the final version of our paper.

---

> > ### Comment · Reviewer_st1h · 2023-08-13
> >
> > Thank you for the detailed responses!

---

### Official Review · Reviewer_ksE2 · 2023-07-06

**Soundness:** 3 good
**Presentation:** 3 good
**Contribution:** 3 good
**Rating:** 7
**Confidence:** 4

**Summary:**

This paper presents SwiFT (Swin 4D fMRI Transformer), a novel Swin Transformer architecture designed for modeling spatiotemporal brain dynamics from high-dimensional 4D functional MRI data. The architecture incorporates a 4D window multi-head self-attention mechanism and absolute positional embeddings, making memory usage and computation efficient. SwiFT outperforms recent state-of-the-art models in several tasks, including predicting sex, age, and cognitive intelligence, based on the evaluation on multiple largest-scale human functional brain imaging datasets. The paper further demonstrates the feasibility of self-supervised pre-training of SwiFT using contrastive loss for improved performance on downstream tasks, marking the first end-to-end learning application of Swin Transformer architecture on dimensional spatiotemporal brain functional data.

**Strengths:**

1. The ability to utilize pretraining technology and leverage large datasets to aid small datasets is a considerable strength of this paper. This is particularly beneficial for fMRI analysis, where large public datasets and smaller private datasets are common. The paper's approach could help mitigate the challenges of small sample sizes.
2. The paper marks an important advancement by directly applying deep learning models to fMRI data. This approach could unify various preprocessing pipelines and simplify the analysis process, which is a significant step forward in the field.
3. The experiments were carried out on three large fMRI datasets, which adds credibility and robustness to the results. By testing their approach on different datasets, the authors ensured that their findings were not limited to a specific dataset, thereby improving the generalizability of the results.

**Weaknesses:**

1. The paper lacks a proper discussion of its limitations. Understanding the constraints of the presented approach is important for future research and application of the study's findings. The authors should consider addressing potential limitations, caveats, and assumptions made in their methodology to provide a more comprehensive view of the work.
2. In Section 4.6, the authors had better thoroughly explain why different patterns emerge when the Input Time Sequence Length varies among different tasks. This lack of in-depth discussion and analysis might hinder the reader's understanding of the method's behavior under different conditions. Therefore, it would be beneficial to elucidate these differences.
3. The discussion in Section 4.4 seems insufficient in terms of relating the study's findings to previous literature. A more detailed comparison with past studies would provide readers with a better understanding of the novelty and contribution of this work. This could include a more explicit discussion of how their findings support previous studies.

**Questions:**

1. Which ROI system is used to preprocess data for these ROI-based methods, like BNT?
2. How do you choose the number of layers?

**Limitations:**

Yes

---

> ### Author Rebuttal · Authors · 2023-08-10
>
> Thank you for the review and raising some important questions. Here is our response:
>
> >*The paper lacks a proper discussion of its limitations...*
> - Although our model has demonstrated high performance and efficiency compared to existing models, it still presents certain limitations for neuroscientists aiming to apply it to their specific subjects. The fMRI data utilized in this experiment ranges from a minimum volume of 383 (586 megabytes) to 1200 (1.3 gigabytes) per subject. While SwiFT significantly reduces the number of parameters and enhances computational speed compared to the existing fMRI Transformer, training a model on such data necessitates more than 24 gigabytes of GPU resources and storage space for thousands of fMRI images. This can pose a significant challenge for researchers with limited computing resources.
> - Our study is based on a sliding window approach and learning is performed on sub-sequences. This only offers a limited description of its capability to handle long-term dynamics. Processing entire fMRI volumes, which can amount to several gigabytes for multiple subjects, is unfeasible considering limitations in GPU resources. When using a sliding window, the model primarily focuses on the local temporal dynamics of the fMRI, which restricts its ability to learn long-term temporal patterns. Hence, when reporting performance, it is essential to conduct additional validation on the uniformity of the distribution of logit values within sub-sequences for each subject.
>
> > *In Section 4.6, the authors had better thoroughly explain why different patterns emerge ...*
>
> - We apologize for not elaborating as to why this behavior might have emerged. Please refer to the general response for our analysis on the matter.
>
> > *The discussion in Section 4.4 seems insufficient...*
> - Across all age groups, we observed brain regions associated with the default mode network, such as the medial prefrontal gyrus (mPFC), posterior cingulate cortex (PCC), precuneus (PCu), and parietal gyrus. Sex difference in these regions were implicated in the literature ([1], [2], [3]). These regions are known to exhibit concurrent activation when not engaged in specific tasks, where strong functional connection exists between each other.
> - Notably, we found different brain contributors to sex classification across different age groups. Unique regions were observed in young adults (HCP) and older-aged adults (UKB). In youth (HCP), the observation centered on the thalamus and insular cortex. On the other hand, among the older adults (UKB), the focus shifted to the inferior temporal gyrus (ITG) and medial orbitofrontal cortex (mOFC). These regions serve as hubs for various cognitive functions, encompassing multisensory integration (Thalamus), emotional processing (insular cortex), higher-level visual processing (ITG), and decision-making (mOFC), where diverse sensory information is synthesized. This result is in line with the previous findings reporting the sex differences in those regions ([4], [5], [6]).
> - These findings suggest that sex difference is prominently evident in regions characterized by robust connectivity with other areas and involved in substantial information exchange. Moreover, we found unique brain contributors to sex across different age groups.
>
> >*Which ROI system is used to preprocess data for these ROI-based methods, like BNT?*
>
> - As mentioned in **Datasets** of section 4.1 on page 6, we have utilized the HCP MMP1 atlas to preprocess our data for ROI-based methods.
>
> >*How do you choose the number of layers?*
>
> - During our earlier phase of testing, starting with the number of layers suggested by the first Swin Transformer paper ([19] of our paper), we have toyed with different configurations by adding or subtracting some layers, or even omitting the last stage. However we discovered that the original number of layers performed the best in general and decided to keep it while shifting our focus to other hyperparameters. As for the effect of other hyperparameters, please refer to our response to the question from reviewer pTkQ.
>
> [1] Ficek-Tani, B., Horien, C., Ju, S., Xu, W., Li, N., Lacadie, C., ... & Fredericks, C. (2023). Sex differences in default mode network connectivity in healthy aging adults. Cerebral Cortex, 33(10), 6139-6151.
>
> [2] Ernst, M., Benson, B., Artiges, E., Gorka, A. X., Lemaitre, H., Lago, T., ... & Martinot, J. L. (2019). Pubertal maturation and sex effects on the default-mode network connectivity implicated in mood dysregulation. Translational psychiatry, 9(1), 103.
>
> [3] Weis, S., Patil, K. R., Hoffstaedter, F., Nostro, A., Yeo, B. T., & Eickhoff, S. B. (2020). Sex classification by resting state brain connectivity. Cerebral cortex, 30(2), 824-835.
>
> [4] Wu, X., Lu, X., Zhang, H., Bi, Y., Gu, R., Kong, Y., & Hu, L. (2023). Sex difference in trait empathy is encoded in the human anterior insula. Cerebral Cortex, 33(9), 5055-5065.
>
> [5] Leming, M., & Suckling, J. (2021). Deep learning for sex classification in resting-state and task functional brain networks from the UK Biobank. NeuroImage, 241, 118409.
>
> [6] Weis, S., Patil, K. R., Hoffstaedter, F., Nostro, A., Yeo, B. T., & Eickhoff, S. B. (2020). Sex classification by resting state brain connectivity. Cerebral cortex, 30(2), 824-835.

---

> > ### Comment · Reviewer_ksE2 · 2023-08-14
> >
> > Thanks for the detailed response. I think providing the open-source implementation can improve this work's impact.

---

> > ### Comment · Reviewer_ksE2 · 2023-08-21
> >
> > After reading the reply, I decide to raise my rating.

---

### Official Review · Reviewer_JwtU · 2023-07-06

**Soundness:** 3 good
**Presentation:** 3 good
**Contribution:** 2 fair
**Rating:** 6
**Confidence:** 4

**Summary:**

The paper focuses on the expansion of the Swin Transformer to a 4D version, enabling it to be trained on 4D fMRI data in an end-to-end manner. Specifically, the proposed method commences by constructing an absolute positional embedding layer across spatially neighboring patches, prior to implementing deep 4D Swin Transformer blocks. To facilitate interaction between the windows, a shifted window MSA technique is employed. The paper also explores the self-supervised pre-training of the method, leveraging instance contrastive loss and local-local temporal contrastive loss. Ultimately, the manuscript presents an evaluation of the method on classification and regression tasks, and conducts an ablation study on the model's efficiency in comparison to the TFF method.

**Strengths:**

The manuscript is coherently presented, and the concept of adapting the Swin Transformer for representation learning using 4D fMRI data is interesting.

The paper's advancement into self-supervised learning, employing both local-local and instance contrastive learning, effectively demonstrates that the proposed SwiFT model exhibits robust generalization capabilities in the context of downstream tasks.

**Weaknesses:**

Swin Transformers primarily restrict self-attention computations to specific sub-windows, which potentially curtails the model's ability to capture information from brain regions that are spatially distant from one another.

The computational cost associated with the proposed method is substantial. Instead of merely contrasting it with TFF, wouldn't a more comprehensive ablation study on the computational cost, compared to other state-of-the-art methods, offer a broader perspective?

The method's experimental evaluation appears restricted (please refer to the questions section for further details).

**Questions:**

The method's evaluation is currently confined to classification and regression. What about what other real-world applications (that might be feasible with the proposed method)? For instance, could there be potential for extending the method to tasks like brain segmentation or image reconstruction?

Could you possibly expand on the concept of integrating 206 spatially neighboring patches into a token? Might it not be more efficient to consider alternative techniques such as tokenization based on brain anatomical regions?

As for the issue of sex classification, the method either failed to surpass baseline methodologies or the difference in accuracy was negligible. Could you delve deeper into potential reasons for this constraint within the study?

**Limitations:**

Yes, the paper covers some limitations of the work.

---

> ### Author Rebuttal · Authors · 2023-08-10
>
> Thank you for the review and providing us with helpful suggestions. Here is our response to some of the questions and concerns raised:
>
> > *Swin Transformers primarily restrict self-attention computations to specific sub-windows...*
>
> - We agree that it is important to address the issue. Please refer to the general rebuttal for our analysis on the matter.
>
> > *...wouldn't a more comprehensive ablation study on the computational cost ... offer a broader perspective?*
>
> - We could compare the computational cost our model against other baseline ROI-based models, although the comparison would be heavily favored towards the ROI-based models. If we compare the computation cost of the model given the input from a subject, the ROI-based models would have to only process a $180 \times 180$ input while SwiFT and TFF has to directly process a $96 \times 96 \times 96 \times 1200$ input. We could take account of the extra one-time preprocessing step that ROI-based models require, although even at that point it would be unfair to directly compare the computation cost of those two types of models.
>
> > *... What about what other real-world applications...*
>
> - Thank you for the good suggestion. SwiFT can be extended to solve many important psychological and neuroscientific problems. Let us introduce three feasible applications for SwiFT:
>     -	Predicting various task-related brain activity from resting-state functional connectivity is a well-established task that has received much attention [1]. The task holds a substantial potential value, particularly for patients or children who encounter challenges performing complicated tasks within fMRI. With SwiFT demonstrating exceptional predictive performance, we anticipate an improved ability to forecast task-related brain activity using the raw resting-state BOLD signal in comparison to functional connectivity methods.
>     -	SwiFT can also be utilized to execute functional segmentation. Previous researchers have extracted spatially independent components from fMRI based on multivariate decomposition methods such as independent component analysis (ICA). Coherent functional brain networks can be discovered based on the relationship between these spatial components. By analyzing the brain dynamics present in resting-state fMRI in a non-linear manner, SwiFT is expected to generate higher quality component maps than existing methods.
>     -	SwiFT can also be extended to brain decoding tasks, where information about what a subject sees or hears is reconstructed from the fMRI. Recent decoding studies have shown that it is possible to predict fMRI activity levels in specific brain regions, such as the visual cortex or inferior temporal gyrus, from the features of words learned by a large language model. By utilizing task fMRIs, we can extend the scope of brain decoding to the entire brain and understand how whole brain regions relate to the visual cortex and external stimuli.
>
> > *... consider alternative techniques such as tokenization based on brain anatomical regions?*
>
> -	Thank you for the intriguing suggestion. To clarify, during the initial patch embedding step, 6×6×6=216 neighboring voxels within a patch are embedded into a 36-dimensional token, which is then used as the input for the Transformer. Incorporating information from the brain anatomical regions could enhance the model's ability to effectively learn the brain's structure. The ROI-based tokenization scheme could also help us reduce the number of tokens and make our model more efficient.
> -	However, the reason for our current approach is that ROI-based tokenization based on a specific population-based template image, such as an atlas, may not reflect the individual specificity of the subject, such as age or race. If we ignore these characteristics and define ROI based on a population-based atlas, we may introduce bias into model training. However, if we differentiate atlases based on the demographic attributes of the data, we become constrained to training distinct models for varying atlases. This may limit the ability to generalize data from multiple ages and races in a single model. Therefore, we adopted a simple yet consistent method for SwiFT, which may also be one argument for using a end-to-end model that does not rely on hand-crafted features.
>
> > *As for the issue of sex classification, the method either failed to surpass baseline methodologies...*
>
> - While for each dataset there are baseline models that show comparable results against our model, on a whole there is no baseline model that consistently performs on all three datasets. The ROI-based models perform well on the ABCD dataset but underperforms on the HCP and UKB dataset. The TFF model performs well on the HCP and UKB dataset but underperforms on the ABCD dataset.
> - For the HCP and UKB dataset, the best performing baseline model (TFF) already performs extremely well, with 0.980 and 0.998 AUC for the HCP and UKB dataset respectively. This does not leave much room for improvement, and may be the reason why our model was only able to have minor gains compared to TFF.
> - As for the ABCD dataset, the ROI-based baseline models (BrainNetCNN, VanillaTF, BNT) have showed comparable or slightly better performance compared to SwiFT, although they have underperformed in other datasets. This may be the result of our choice of brain atlas for these models was appropriate for the ABCD dataset but not for the other datasets. We argue that this is one inherent problem of relying on such hand-crafted features, since we have to make sure the application of such feature is appropriate on a case-by-case basis.
>
> [1] Tavor, I. et al. (2016). Task-free MRI predicts individual differences in brain activity during task performance. Science, 352(6282), 216-220.
>
> [2] Hatamizadeh, A. et al. (2021). Swin unetr: Swin transformers for semantic segmentation of brain tumors in mri images. In International MICCAI Brainlesion Workshop (pp. 272-284).

---

> > ### Comment · Reviewer_JwtU · 2023-08-20
> >
> > Thank you for the responses. After reviewing the author's answers, I've increased my score to "weak accept." However, I still believe that some aspects of the model's evaluation are incremental when compared to other methods. Additionally, if certain datasets are easy to address, the evaluation could be reframed to facilitate a more robust comparison.

---

### Official Review · Reviewer_pTkQ · 2023-07-07

**Soundness:** 3 good
**Presentation:** 3 good
**Contribution:** 2 fair
**Rating:** 6
**Confidence:** 5

**Summary:**

* The authors seek to develop an approach for modeling spatiotemporal brain dynamics, as measured with resting state functional MRI. For this, they have extended a transformer-based architecture to handle time-varying 3D scan data.
* They seek to predict subject traits and attributes like age, sex, and cognitive scores from the learned representations using three publicly available datasets.
* They have succeeded in outperforming state-of-the-art models in these tasks while reducing the computational complexity and increasing throughput in model evaluations.
* Additionally, they have demonstrated the usefulness of pre-training such models on downstream tasks.
* Lastly, the relationships captured by the model between the subject attributes and the brain regions are consistent with prior work.



**Strengths:**

* The problem statement is well-motivated, and the limitations of prior works (i.e., ROI-based and two-step approaches) are detailed.
* The results were replicated across three datasets and various tasks, showcasing this architecture's robustness.
* The authors show the learned features are general enough to help downstream tasks through pre-training.
* The authors have performed computational complexity analysis. Because their model has almost double the throughput compared to the second-best-performing model, it paves the way to use these models in real-time settings.
* Regions of the brain used by the model (as in Section 4.4) to predict age align with previous work, which increases confidence that the model is learning meaningful features.
* The authors have submitted their full code, which assists replicability and helps the scientific community extend their work.

**Weaknesses:**

* The novelty of the architecture vis-a-vis latest advances in transformer architectures ([23], [34], [35]) seems limited.
* The authors aimed to learn representations for brain dynamics but focussed only on predicting traits (age, sex etc), because they used only resting state scans. Dynamic attributes such as task-level performance measures -- e.g., reaction times and accuracies in HCP tasks), cognitive load in a working memory task etc -- are relevant to establish the generality of the findings. The model should be tested on these tasks also.
* While the authors have detailed the approach for the two contrastive losses if they could justify the choice of positive and negative samples through domain knowledge or established methods, it would help increase the confidence in the proposed approach.
* The authors have mentioned performing ablation studies to "substantiate their modeling choices." I expect that this includes studying the effect of hyperparameters like the number of layers, channel size, window size, etc., given that this is a novel architecture. Details of such studies should have been mentioned.
* Additionally, it would be helpful to see the loss curves of models for different datasets and targets. It will help understand the choice of a low number of epochs for training various models.

**Questions:**

* Could the authors clarify if the parameters for z-scoring, i.e., mean and variance, were learned over the whole data or only on the training data (& applied to validation and test data)?
* I would expect that using the whole fMRI time series for any task prediction would provide better results than using sub-sequences.  Could the authors provide more intuition about the seemingly counterintuitive results obtained in Section 4.6, especially in UKB (e.g., age prediction)?
* While we commend that the authors have shared their code, it would be helpful if they could also share their trained models (especially due to the high compute power required to generate these results). It would help the scientific community at large and also be in line with the spirit of Section 4.3.

**Limitations:**

* Explanation methods like IG are generally brittle. It would be helpful if one more explanation method could confirm the results in Section 4.4
* While a shifting window enables efficient feature extraction, it limits model expressivity. It would be useful to know how to extract features that span large spatial/temporal ranges. This could be particularly relevant for task-based fMRI data since signals of varying timescales could affect behavior in such paradigms.
* Authors have mentioned that they divided the fMRI data into sub-sequences due to memory constraints. Moreover, as the authors have used four Nvidia A100s (a total of 160GB of GPU memory with NvLink support), they are already on the higher end of compute power. It would be helpful to understand what those memory constraints were and suggestions, if any, on tackling them, as this will have implications in replicating and/or extending their work.

---

> ### Author Rebuttal · Authors · 2023-08-09
>
> Thank you for raising some important questions and providing us with helpful feedback. Here is our response to the questions and concerns raised:
>
> >*The novelty of the architecture ...*
>
> - The novelty of our architecture may seem to be limited, however there are several points that we claim to be novel.
> - As previous works (e.g., [23] in the manuscript) have extended the Swin Transformer to accept 3D inputs, we extend it one step further to directly process 4D inputs. This comes with its own set of challenges, such as how to deal with the patch merging step combining larger number of tokens or the memory and computation constraints due to the added dimension.
> - VAT ([35] of our paper) technically utilizes the Swin Transformer to process 4D inputs, although the Swin Transformer module was used to process pair of intermediate 2D feature maps instead of a "true" 4D data. The Swin Transformer is also missing some key features such as the patch merging step and the hierarchical structure. Due to these distinctions we believe that our introduction of a fully-fledged 4D Swin Transformer architecture can still be considered as a novel contribution.
>
> >*...The model should be tested on these tasks also.*
> - Thank you for the suggestion. As briefly mentioned in the general rebuttal regarding long-range interactions, we also believe this to be an exciting future work. We expect SwiFT to perform well on these datasets compared to ROI-based models since SwiFT does not "delete" the temporal information during the ROI preprocessing step.
>
> >*...approach for the two contrastive losses...*
>
> -  We will add the following to the final version of the paper.
> - Our choices are based on an established method called TCLR ([38] of our paper). We adapt two loss functions from here to help SwiFT achieve a better understanding of temporal dynamics by a) **distinguishing fMRI scans from different subjects** and b) **distinguishing fMRI scans from different timestamps from the same subject**.
> - The instance contrastive loss accomplishes a) by considering fMRI scans from the same subject as positive pairs and fMRI scans from a different subject as negative pairs.
> - The local-local contrastive loss accomplishes b) by considering different augmentations of the same scan as positive pairs and fMRI scans from different timestamps as negative pairs.
> - We hope this clarifies the use of our contrastive loss.
>
> > *... performing ablation studies to "substantiate their modeling choices." ...*
>
> - Although we had originally intended the phrase "substantiate our modeling choices" to refer to our qualitative design choices such as the switch to the absolute position embedding scheme, we understand the reviewer's concerns.
> - Here are some of the test results on the HCP sex classification task. Tweaking the hyperparameters had surprisingly little effect, and after toying on a larger ABCD dataset, we settled on the architecture mentioned in the paper while balancing the model's performance, efficiency, and memory usage.
>
> | Temporal Window Size $P$ |4| 2 | 6 | 4 | 4 |
> |-|-|-|-|-|-|
> | **Channel Number $C$**|**36**|**36**|**36**|**24**|**48**|
> |Accuracy (%)|$92.9\pm1.51$|$93.2\pm2.18$|$93.2\pm3.22$|$94.1\pm2.41$|$92.7\pm1.76$|
> |AUC (%)|$98.0\pm1.79$|$97.9\pm1.40$|$98.0\pm1.65$|$98.6\pm0.6$|$97.2\pm1.1$|
>
> >*Additionally, it would be helpful to see the loss curves of models...*
>
> - Thank you for the suggestion, and we have included the loss curves in the general rebuttal PDF (figure 1). We would be happy to share the details in our final version of the paper.
>
> >*Could the authors clarify if the parameters for z-scoring...*
>
> - We originally utilized the mean and variance from the whole data to normalize the regression target labels, but we acknowledge the err in our methodology and modified our normalization to only use the mean and variance from the training data for each split.
> - Even though in practice this should not impact the test results since the same normalization method was used for all baseline models, and the change of the mean and variance due to this switch is minimal (table 1 of PDF), we trained our model using the correct normalization method and posted the training curve to make sure the training process was not impacted due to this change (figure 2 of PDF). We will make sure to update the regression results using the correct normalization method in the final version of our paper.
>
> > *I would expect that using the whole fMRI time series ...*
>
> - We agree that in an ideal scenario where we have access to a very large dataset the reviewer's expectation should be valid, however we believe that the input time sequence length should be considered as a hyperparameter in more limited settings. Please refer to the general rebuttal response for our opinion regarding the matter.
>
> > *...it would be helpful if they could also share their trained models...*
>
> - We agree with the sentiment and would be happy to release our trained models and open-source code alongside the final draft of our paper.
>
> >*Explanation methods like IG are generally brittle...*
>
> - The interpretation method we used for our analysis, which is Integrated gradient with Smoothgrad sQuare (IG-SQ), is known to be more robust [1] thanks to noise tunneling and the squaring of estimates. We believed this interpretation method to be the most appropriate for our model, and we would be happy to try other methods if there are any suggestions.
>
> > *... a shifting window ... limits model expressivity...*
>
> - We understand the concerns, and please refer to the general rebuttal regarding our analysis of the matter.
>
> > *Authors have mentioned that they divided the fMRI data into sub-sequences...*
>
> - We apologize for the confusion. Please refer to our general rebuttal discussing our computing setup.
>
> [1] Hooker, Sara, et al. "A benchmark for interpretability methods in deep neural networks." Advances in neural information processing systems 32 (2019).

---

> > ### Comment · Reviewer_pTkQ · 2023-08-20
> >
> > I thank the authors for the clarifications and the detailed responses. Validation loss trends are somewhat irregular for the Intelligence score prediction (e.g. UKB, ABCD). Although the method is cutting-edge, demonstrating more fine-grained applications, such as predicting within-scanner scores, is necessary to take it to the next level of impact.

---

> > > ### Author Response · Authors · 2023-08-21
> > >
> > > Thanks for your great feedback. Regarding the irregular loss trends in ABCD intelligence prediction, it's noteworthy that these fluctuations in loss occur during epochs in which the learning rate increases within the learning rate scheduler (Cosine Annealing with Warmup restarts). In UKB intelligence prediction, we observed that the model's performance tended to converge at the early stage of the training. To avoid overfitting,  we early stopped the training. The fine-grained applications for SwiFT you mentioned will be thoroughly addressed in our forthcoming research endeavors.

---

### Official Review · Reviewer_bLJJ · 2023-07-10

**Soundness:** 3 good
**Presentation:** 3 good
**Contribution:** 3 good
**Rating:** 6
**Confidence:** 4

**Summary:**

This paper introduces SwiFT, a novel Swin Transformer-based architecture for analyzing high-dimensional functional brain MRI data. SwiFT learns end-to-end to process spatiotemporal brain functional data, and it achieves state-of-the-art performance on several tasks, including sex classification, age prediction, and cognitive intelligence prediction. The authors also demonstrate the feasibility of applying the pre-train and fine-tune framework to SwiFT, empowering researchers to construct large-scale foundation models for fMRI.

**Strengths:**

- The paper propose Swin Transformer-based architecture that can learn brain dynamics directly from 4D functional brain MRI data in an end-to-end fashion, encouraging researchers to construct foundation models for fMRI.
- The authors provide a clear and concise introduction to the problem of analyzing high-dimensional functional brain MRI data and explain the motivation for developing SwiFT.
- Results reveal that SwiFT consistently outperforms recent state-of-the-art models. The authors also conduct ablation studies to substantiate the modeling choices and present interpretation results using IG-SQ for SwiFT's predictions.
- SwiFT has the potential to facilitate scalable learning of functional brain imaging in neuroscience research by reducing the hurdles associated with applying Transformer models to high-dimensional fMRI.
- The authors also provide interpretability of SwiFT's predictions, which can help researchers better understand the brain's spatiotemporal dynamics.

**Weaknesses:**

- The paper compares SwiFT with limited recent state-of-the-art models,. It would be interesting to see how SwiFT compares with more types of models, such as CNNs or GNNs, which are commonly used in fMRI analysis.
- While the paper presents interpretation results using IG-SQ for SwiFT's predictions, it would be helpful to have a more in-depth discussion and how they can be used to better understand the brain's spatiotemporal dynamics.
- It would be helpful to have a more in-depth discussion of the limitations of SwiFT, such as its sensitivity to noise in the data or its ability to generalize to new datasets.
- The paper does not provide open-source code for SwiFT, which could limit its adoption and reproducibility.

**Questions:**

- Could the authors provide more details on the computational resources required to train and evaluate SwiFT? This would help to better understand the practical feasibility of using SwiFT in real-world applications.
- Can the authors provide more details on the implementation of the 4D window multi-head self-attention mechanism and absolute positional embeddings used in SwiFT? This would help to better understand the specifics of SwiFT's architecture and how it differs from other Transformer-based models.
- Could the authors provide a more in-depth discussion of the limitations of SwiFT, such as its sensitivity to noise in the data or its ability to generalize to new datasets?

**Limitations:**

Consider the potential impact of SwiFT on the broader field of neuroscience and medicine, such as its potential to facilitate scalable learning of functional brain imaging. This would help to better understand the potential benefits and drawbacks of using SwiFT in research and clinical settings.

---

> ### Author Rebuttal · Authors · 2023-08-10
>
> Thank you for the detailed analysis of our work and for the insightful suggestions. Here we address some of the questions and concerns raised:
>
> > *...It would be interesting to see how SwiFT compares with more types of models...*
>
> -  Thank you for providing us with some pointers. To faithfully evaluate the effectiveness of our model, we have already reproduced multiple models as baselines. One of such models is BNT ([12] of our paper), the current state-of-the-art model to the best of our knowledge. BNT improves upon the traditional GNN-based approaches for fMRI analysis and represents the cutting-edge method of similar GNN-based approaches.
>
> > *While the paper presents interpretation results ...*
> - Thank you for your inquiry. Currently, we employ a method of calculating IG-SQ for each sub-sequence and subsequently averaging along the time axis. Through this approach, we have identified which regions, on average, exhibit the highest explanatory power for gender differences.
> - For these IG-SQ values to hold further significance, conducting additional analyses on their relationship with brain connectivity is necessary. The brain regions showing high IG-SQ values in our findings, such as the thalamus, insular cortex, and inferior temporal gyrus, are primarily hubs for integrating various sensory information and exhibit connectivity with several other brain regions. A crucial research question is investigating whether this high connectivity contributes to the elevated IG-SQ values.
> - Furthermore, exploring how well IG-SQ values can account for individual differences is worthwhile. IG-SQ values serve as a more direct measure of sex difference. Each individual possesses unique IG-SQ values, which might also represent individual differences. Consequently, investigating the predictive capacity of Interpretation maps for mental disorders characterized by significant gender differences, such as depression or ASD, is a crucial research avenue in the future.
>
> > *The paper does not provide open-source code ...*
>
> -  We apologize that we had to provide our code only as a file in the supplementary material due to the anonymous review process. We would be happy to release our open-source code alongside the final draft of our paper.
>
> > *Could the authors provide more details on the computational resources ...*
>
> -  Please refer to the general rebuttal for the additional details.
>
> > *Can the authors provide more details on the implementation...*
> - Thank you for the comment. We supplement our implementation details with the following:
> - The 4D window multi-head self-attention (4DW-MSA) mechanism is an extension of its 2D variant from the Swin Transformer. The core idea of the windowed attention mechanism is that tokens only attend to tokens that are spatially or temporally adjacent to them. This would allow the Transformer to achieve lower computation and memory complexity.
> - To be specific, we divide the entire 4D space into smaller sub-spaces (windows), and tokens within a sub-space only attend to tokens within the same sub-space. Additionally, after each layer the windows are shifted by half of its width to allow the tokens to exchange information with new adjacent tokens.
> - The absolute positional embeddings, although not commonly used, were introduced in our work to improve the model's efficiency. The absolute positional embeddings have a simple premise: we add a (learnable) embedding vector to each of the tokens based on their position. This would be adding a unique vector for each of the 81,260 possible positions for our configuration. As a design choice, we separate the time dimension and space dimension and employ 20 time-embedding vectors and $16 \times 16 \times 16 = 4,096$ space-embedding vectors and add them separately.
>
>
> > *a more in-depth discussion of the limitations of SwiFT...*
> - A limitation of SwiFT is its inability to process entire sequence as a whole, necessitating the division into sub-sequences. We theorize it may not be an entirely negative aspect, however it would be difficult to test out processing entire sequences due to memory constraints. For a more comprehensive explanation, please refer to the concept on the general rebuttal.
> - For limitations associated with generalizing to new datasets, please refer to the response provided below.
>
> > *Consider the potential impact of SwiFT on the broader field ...*
>
> -	Thank you for your interest in the potential impact of SwiFT. We expect SwiFT to become a foundation model for functional brain imaging that can solve various problems in neuroscience and medicine. In recent years, it has been a trend to train large-scale models and then perform new tasks through few-shot learning. However, functional imaging has not had such an approach due to the lack of models and computational resources for large-scale training, and we believe that SwiFT can be an effective solution.
> -	In research and clinical settings, SwiFT can be used to solve specific, well-defined problems. For instance, SwiFT could be directly employed for the diagnosis and prognosis of diseases, and the features learned by SwiFT could be harnessed for subtyping heterogeneous psychiatric disorders. However, owing to the intrinsic nature of transformers, the efficacy of SwiFT might not be fully realized with smaller datasets, and it becomes necessary to fine-tune the model using weights pre-trained on larger datasets.
> -	In our work it has been observed that transfer learning exhibits restricted enhancements in specific tasks. However, this limitation can be due to conducting pre-training solely on a single type of data. We anticipate that broadening the scope of pre-training to encompass diverse datasets, such as task fMRI, will augment the effectiveness of transfer learning. This stands as a crucial avenue for future inquiries.

---

> ### Comment · Reviewer_bLJJ · 2023-08-21
> **Response to Rebuttal**
>
> Thanks the authors for the explanation!

---

### Author Rebuttal · Authors · 2023-08-10


We thank the reviewers for taking their time to review our work and provide constructive feedback that would help us improve our work. Here are our responses to common questions and concerns raised by multiple reviewers.

### Our computational resources (for reviewer bLJJ, pTkQ)

- We have used four NVIDIA A100 GPUs with a distributed data-parallel (DDP) strategy. Using multiple GPUs speeds up the training process by processing subjects in parallel. However, the number of samples (and the amount of memory each GPU can use) each GPU can process does not change as the number of GPUs increases.
- For training, on a single NVIDIA A100 GPU, it takes about 40 minutes for each epoch to train on the HCP dataset,composed of 1,084 subjects each with \~1,200 volumes of ($96\times96\times96$) fMRI scans, totaling \~6 hours to complete training (e.g., sex classification). The GPU profiling tool showed that the GPU was using 17.6 GB of memory during our typical training session with a mini-batch size of 8. Although we have mainly used a GPU with a 40GB memory, we have confirmed that NVIDIA A5000 GPU with a 24GB memory is enough to train our model. Using a DDP strategy with 4 GPUs speeds up the training by a factor of 3\~4, provided that the input data can be loaded from the storage into the GPUs fast enough.
- For evaluation, using the throughput measured on a single NVIDIA A100 GPU (Table 2), the model can process the 1200 frames of a single subject in 0.57 seconds. With a lower-powered A5000 GPU we could still process a subject under a second.
- We would be happpy to add these additional details to the final version of our paper.

### Swin Transformer's ability to capture long-range interactions (for reviewer pTkQ, JwtU)
- As reviewers pTkQ and JwtU have pointed out, owing to the local nature of the attention mechanism, our model might not capture long-range interactions between spatially/temporally separated tokens. We understand the reviewer's concerns, but we believe our model have some built-in mechanisms to learn about long-range communication between tokens. Here we would like to explain how our model may learn about long-range relationships.
- On the spatial dimensions, the Swin Transformer model can cover large ranges on the later parts of its layers through the patch merging step. On our configuration, after the initial patch embedding step where $6 \times 6 \times 6 = 216$ neighboring voxels are embedded into a patch, a $4 \times 4 \times 4$ window can cover a range equivalent to $24 \times 24 \times 24$ voxels from our $96 \times 96 \times 96$ input. However, after the patch merging step where 8 neighboring tokens are merged, the same window can cover a range equivalent to $48 \times 48 \times 48$ voxels. After another patch merging step, on stage 3, this grows to $96 \times 96 \times 96$ which is already the same size as our entire input. From this point on, a single window covers the entire span of our input in the spatial dimension, thanks to the patch merging step.
- On the temporal dimension, where patch merging does not occur on the current version of the model, we needed another method that can ensure long-range interation between tokens. The shifting window allows a token to interact with tokens that are ~10 time frames away with our setup, but to allow longer range interactions we made the model do a full global attention on the final stage as mentioned on section 3.1. Thus, the model processes the entire temporal range on the final stage, allowing it to introduce long-range temporal interactions in the end. We acknowledge that there could be more effective methods to allow long-range interactions, and it would be an exciting problem to tackle when we are dealing with more temporally dynamic datasets as reviewer pTkQ have metioned.

### Limitations regarding the utilization of sub-sequences (section 4.6) (for all reviewers)
- In our setup, instead of inputting the entire fMRI sequence of a subject all at once, we have opted to divide it into 20-frame sub-sequence and use the divided sub-sequences as an input to our model. This was mainly due to memory constraints and to regularize the number of input tokens to our model. We have investigated the effect of changing the length of this input sub-sequence in section 4.6. Here, we would like to elaborate as to why the optimal length seems to change for each dataset and task.
- Inputting a longer time sequence into the model would allow the model to attend to longer sequences at once, at the expense of bloating the model with more parameters and causing it to overfit. Therefore in a setting where we are dealing with a limited number of training data, using a shorter sub-sequece has a possibility to be beneficial to the model's generalization capabilities.
- We also note that the total amount of information used for our predictions remains the same regardless of the input time sequence length, as the averaged logits from all sub-sequences of the subject is used for the prediction.
- The datasets used for this work are resting-state fMRI scans, where the image does not change drastically over time. In the future, it would be interesting to investigate the effect of input time sequence length on more temporally dynamic datasets, such as task-based fMRI scans.

---

### Decision · Program_Chairs · 2023-09-21

**Decision:**

Accept (poster)

**Comment:**

This paper introduces SwiFT, a novel Swin Transformer-based architecture for analyzing high-dimensional functional brain MRI data. The reviewers unanimously support the acceptance of the paper, based on that novel application with end-to-end learning approach, as well as strong empirical results. However, there are some limitations pointed out by the reviewers such as limited comparison to some other model types like CNNs and GNNs, high computational resources required for training. Overall, it is a good paper.